# What Makes for Good Views for Contrastive Learning?

**Yonglong Tian**
MIT CSAIL

**Chen Sun**
Google, Brown University

**Ben Poole**
Google Research

**Dilip Krishnan**
Google Research

**Cordelia Schmid**
Google Research

**Phillip Isola**
MIT CSAIL

## Abstract

Contrastive learning between multiple views of the data has recently achieved state of the art performance in the field of self-supervised representation learning. Despite its success, the influence of different view choices has been less studied. In this paper, we use theoretical and empirical analysis to better understand the importance of view selection, and argue that we should reduce the mutual information (MI) between views while keeping task-relevant information intact. To verify this hypothesis, we devise unsupervised and semi-supervised frameworks that learn effective views by aiming to reduce their MI. We also consider data augmentation as a way to reduce MI, and show that increasing data augmentation indeed leads to decreasing MI and improves downstream classification accuracy. As a by-product, we achieve a new state-of-the-art accuracy on unsupervised pre-training for ImageNet classification (73% top-1 linear readout with a ResNet-50)[1].

## 1 Introduction

It is commonsense that how you look at an object does not change its identity. Nonetheless, Jorge Luis Borges imagined the alternative. In his short story on *Funes the Memorious*, the titular character becomes bothered that a "dog at three fourteen (seen from the side) should have the same name as the dog at three fifteen (seen from the front)" [6]. The curse of Funes is that he has a perfect memory, and every new way he looks at the world reveals a percept minutely distinct from anything he has seen before. He cannot collate the disparate experiences.

Most of us, fortunately, do not suffer from this curse. We build mental representations of identity that discard *nuisances* like time of day and viewing angle. The ability to build up *view-invariant* representations is central to a rich body of research on multiview learning. These methods seek representations of the world that are invariant to a family of viewing conditions. Currently, a popular paradigm is contrastive multiview learning, where two views of the same scene are brought together in representation space, and two views of different scenes are pushed apart.

This is a natural and powerful idea but it leaves open an important question: "which viewing conditions should we be invariant to?" It's possible to go too far: if our task is to classify the time of day then we certainly should not use a representation that is invariant to time. Or, like Funes, we could go not far enough: representing each specific viewing angle independently would cripple our ability to track a dog as it moves about a scene.

We therefore seek representations with enough invariance to be robust to inconsequential variations but not so much as to discard information required by downstream tasks. In contrastive learning,

the choice of "views" is what controls the information the representation captures, as the framework results in representations that focus on the shared information between views [42]. Views are commonly different sensory signals, like photos and sounds [3], or different image channels [53] or slices in time [55], but may also be different "augmented" versions of the same data tensor [7]. If the shared information is small, then the learned representation can discard more information about the input and achieve a greater degree of invariance against nuisance variables. How can we find the right balance of views that share just the information we need, no more and no less?

We investigate this question in two ways: 1) we demonstrate that the optimal choice of views depends critically on the downstream task. If you know the task, it is often possible to design effective views. 2) We empirically demonstrate that for many common ways of generating views, there is a sweet spot in terms of downstream performance where the mutual information (MI) between views is neither too high nor too low.

Our analysis suggests an "InfoMin principle". A good set of views are those that share the minimal information necessary to perform well at the downstream task. This idea is related to the idea of minimal sufficient statistics [48] and the Information Bottleneck theory [54, 2], which have been previously articulated in the representation learning literature. This principle also complements the already popular "InfoMax principle" [34] , which states that a goal in representation learning is to capture as much information as possible about the stimulus. We argue that maximizing information is only useful in so far as that information is task-relevant. Beyond that point, learning representations that throw out information about nuisance variables is preferable as it can improve generalization and decrease sample complexity on downstream tasks [48].

Based on our findings, we also introduce a semi-supervised method to *learn* views that are effective for learning good representations when the downstream task is known. We additionally demonstrate that the InfoMin principle can be practically applied by simply seeking stronger data augmentation to further reduce mutual information toward the sweet spot. This effort results in state of the art accuracy on a standard benchmark.

Our contributions include:

- Demonstrating that optimal views for contrastive representation learning are task-dependent.
- Empirically finding a U-shaped relationship between an estimate of mutual information and representation quality in a variety of settings.
- A new semi-supervised method to learn effective views for a given task.
- Applying our understanding to achieve state of the art accuracy of $73.0\%$ on the ImageNet linear readout benchmark with a ResNet-50.

## 2 Related Work

Recently the most competitive methods for learning representations without labels have been self-supervised contrastive representation learning [42, 26, 58, 53, 49, 7]. These methods learn representations by a "contrastive" loss which pushes apart dissimilar data pairs while pulling together similar pairs, an idea similar to exemplar learning [17]. Models based on contrastive losses have significantly outperformed other approaches [64, 30, 43, 53, 16, 41, 15, 19, 62].

One of the major design choices in contrastive learning is how to select the similar (or *positive*) and dissimilar (or *negative*) pairs. The standard approach for generating positive pairs without additional annotations is to create multiple *views* of each datapoint. For example: luminance and chrominance decomposition [53], randomly augmenting an image twice [58, 7, 4, 23, 60, 50, 65, 67], using different time-steps of videos [42, 66, 46, 22, 21], patches of the same image [27, 42, 26], multiple sensory data [39, 9, 44], or text and its context [37, 59, 35, 32]. Negative pairs can be randomly chosen images/videos/texts. Theoretically, we can think of the positive pairs as coming from a joint distribution over views $p(\mathbf{v_1}, \mathbf{v_2})$, and the negative pairs from a product of marginals $p(\mathbf{v_1})p(\mathbf{v_2})$. The contrastive learning objective InfoNCE [42] (or Deep InfoMax [26]) is developed to maximize a lower bound on the mutual information between the two views $I(\mathbf{v_1}; \mathbf{v_2})$. Such connection has been discussed further in [45, 56].

Leveraging labeled data in contrastive representation learning has been shown to guide representations towards task-relevant features that improve performance [61, 25, 29, 57]. Here we use labeled data to

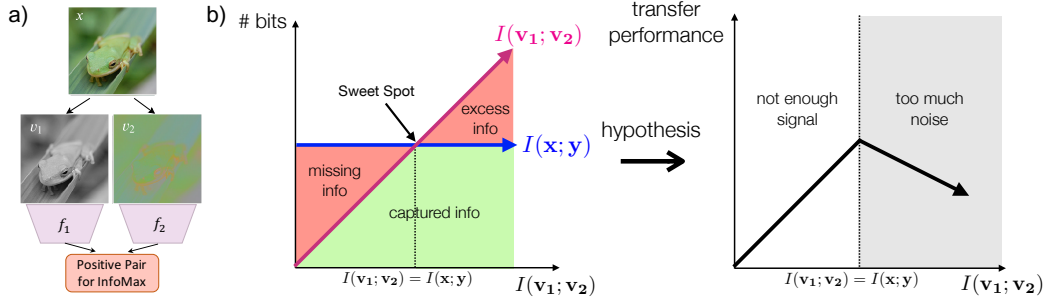

Figure 1: **(a)** Schematic of multiview contrastive representation learning, where an image is split into two views, and passed through two encoders to learn an embedding where the views are close relative to views from other images. **(b)** When we have views that maximize $I(\mathbf{v_1};\mathbf{y})$ and $I(\mathbf{v_2};\mathbf{y})$ (how much task-relevant information is contained) while minimizing $I(\mathbf{v_1};\mathbf{v_2})$ (information shared between views, including both task-relevant and irrelevant information), there are three regimes: *missing information* which leads to degraded performance due to $I(\mathbf{v_1};\mathbf{v_2}) < I(\mathbf{x};\mathbf{y})$; *excess noise* which worsens generalization due to additional noise; *sweet spot* where the only information shared between $\mathbf{v_1}$ and $\mathbf{v_2}$ is task-relevant and such information is complete.

learn better views, but still perform contrastive learning using only unlabeled data. Future work could combine these approaches to leverage labels for both view learning and representation learning.

## 3 What Are the Optimal Views for Contrastive Learning?

In this section, we first introduce the standard multiview contrastive representation learning formulation, and then investigate what would be the optimal views for contrastive learning.

### 3.1 Multiview Contrastive Learning

Given two random variables $\mathbf{v_1}$ and $\mathbf{v_2}$, the goal of contrastive learning is to learn a parametric function to discriminate between samples from the empirical joint distribution $p(\mathbf{v_1})p(\mathbf{v_2}|\mathbf{v_1})$ and samples from the product of marginals $p(\mathbf{v_1})p(\mathbf{v_2})$. The resulting function is an estimator of the mutual information between $\mathbf{v_1}$ and $\mathbf{v_2}$, and the InfoNCE loss [42] has been shown to maximize a lower bound on $I(\mathbf{v_1};\mathbf{v_2})$. In practice, given an anchor point $\mathbf{v_{1,i}}$, the InfoNCE loss is optimized to score the correct positive $\mathbf{v_{2,i}} \sim p(\mathbf{v_2}|\mathbf{v_{1,i}})$ higher compared to a set of $K$ distractors $\mathbf{v_{2,j}} \sim p(\mathbf{v_2})$:

$$\mathcal{L}_{\text{NCE}} = -\mathbb{E}\left[\log \frac{e^{h(\mathbf{v_{1,i}},\mathbf{v_{2,i}})}}{\sum_{j=1}^{K} e^{h(\mathbf{v_{1,i}},\mathbf{v_{2,j}})}}\right] \tag{1}$$

Minimizing this loss equivalently maximizes a lower bound (a.k.a. $I_{\text{NCE}}(\mathbf{v_1};\mathbf{v_2})$) on $I(\mathbf{v_1};\mathbf{v_2})$, i.e., $I(\mathbf{v_1};\mathbf{v_2}) \geq \log(K) - \mathcal{L}_{\text{NCE}} = I_{\text{NCE}}(\mathbf{v_1};\mathbf{v_2})$. In practice, $\mathbf{v_1}$ and $\mathbf{v_2}$ are two views of the data $\mathbf{x}$, such as different augmentations of the same image [58, 4, 23, 8, 7], different image channels [53], or video and text pairs [52, 36, 33]. The score function $h(\cdot,\cdot)$ typically consists of two encoders ($f_1$ for $\mathbf{v_1}$ and $f_2$ for $\mathbf{v_2}$), which may or may not share parameters depending on whether $\mathbf{v_1}$ and $\mathbf{v_2}$ are from the same domain. The resulting representations are $\mathbf{z_1} = f_1(\mathbf{v_1})$ and $\mathbf{z_2} = f_2(\mathbf{v_2})$ (see Fig. 1a).

**Definition 1.** *(Sufficient Encoder) The encoder $f_1$ of $\mathbf{v_1}$ is sufficient in the contrastive learning framework if and only if $I(\mathbf{v_1};\mathbf{v_2}) = I(f_1(\mathbf{v_1});\mathbf{v_2})$.*

Intuitively, the encoder $f_1$ is sufficient if the amount of information in $\mathbf{v_1}$ about $\mathbf{v_2}$ is lossless during the encoding procedure. In other words, $\mathbf{z_1}$ has kept all the information that the contrastive learning objective requires. Symmetrically, $f_2$ is sufficient if $I(\mathbf{v_1};\mathbf{v_2}) = I(\mathbf{v_1};f_2(\mathbf{v_2}))$.

**Definition 2.** *(Minimal Sufficient Encoder) A sufficient encoder $f_1$ of $\mathbf{v_1}$ is minimal if and only if $I(f_1(\mathbf{v_1});\mathbf{v_1}) \leq I(f(\mathbf{v_1});\mathbf{v_1}), \forall f$ that is sufficient.*

Among those encoders which are sufficient, the minimal ones only extract relevant information of the contrastive task and throw away other irrelevant information. This is appealing in cases where the views are constructed in a way that all the information we care about is shared between them.

The representations learned in the contrastive framework are typically used in a separate downstream task. To characterize what representations are good for a downstream task, we define the optimality of representations. To make notation simple, we use $\mathbf{z}$ to mean it can be either $\mathbf{z_1}$ or $\mathbf{z_2}$.

**Definition 3.** *(Optimal Representation of a Task) For a task $\mathcal{T}$ whose goal is to predict a semantic label $\mathbf{y}$ from the input data $\mathbf{x}$, the optimal representation $\mathbf{z}^*$ encoded from $\mathbf{x}$ is the minimal sufficient statistic with respect to $\mathbf{y}$.*

This says a model built on top of $\mathbf{z}^*$ has all the information necessary to predict $\mathbf{y}$ as accurately as if it were to access $\mathbf{x}$. Furthermore, $\mathbf{z}^*$ maintains the smallest complexity, i.e., containing no other information besides that about $\mathbf{y}$, which makes it more generalizable [48]. We refer the reader to [48] for a more in depth discussion about optimal visual representations and minimal sufficient statistics.

### 3.2 Three Regimes of Information Captured

As our representations $\mathbf{z_1}, \mathbf{z_2}$ are built from our views and learned by the contrastive objective with the assumption of minimal sufficient encoders, the amount and type of information shared between $\mathbf{v_1}$ and $\mathbf{v_2}$ (i.e., $I(\mathbf{v_1}; \mathbf{v_2})$) determines how well we perform on downstream tasks. As in information bottleneck [54], we can trace out a tradeoff between how much information our views share about the input, and how well our learned representation performs at predicting $\mathbf{y}$ for a task. Depending on how our views are constructed, we may find that we are keeping too many irrelevant variables while discarding relevant variables, leading to suboptimal performance on the information plane. Alternatively, we can find the views that maximize $I(\mathbf{v_1}; \mathbf{y})$ and $I(\mathbf{v_2}; \mathbf{y})$ (how much information is contained about the task label) while minimizing $I(\mathbf{v_1}; \mathbf{v_2})$ (how much information is shared about the input, including both task-relevant and irrelevant information). Even in the case of these optimal traces, there are three regimes of performance we can consider that are depicted in Fig. 1b, and have been discussed previously in information bottleneck literature [54, 2, 18]:

1. *Missing information*: When $I(\mathbf{v_1}; \mathbf{v_2}) < I(\mathbf{x}; \mathbf{y})$, there is information about the task-relevant variable that is discarded by the view, degrading performance.

2. *Sweet spot*: When $I(\mathbf{v_1}; \mathbf{y}) = I(\mathbf{v_2}; \mathbf{y}) = I(\mathbf{v_1}; \mathbf{v_2}) = I(\mathbf{x}; \mathbf{y})$, the only information shared between $\mathbf{v_1}$ and $\mathbf{v_2}$ is task-relevant, and there is no irrelevant noise.

3. *Excess noise*: As we increase the amount of information shared in the views beyond $I(\mathbf{x}; \mathbf{y})$, we begin to include additional information that is irrelevant for the downstream task. This can lead to worse generalization on the downstream task [2, 47].

We hypothesize that the best performing views will be close to the sweet spot: containing as much task-relevant information while discarding as much irrelevant information in the input as possible. More formally, the following InfoMin proposition articulates which views are optimal supposing that we know the specific downstream task $\mathcal{T}$ in advance. The proof is in Section A.2 of the Appendix.

**Proposition 3.1.** *Suppose $f_1$ and $f_2$ are minimal sufficient encoders. Given a downstream task $\mathcal{T}$ with label $\mathbf{y}$, the optimal views created from the data $\mathbf{x}$ are $(\mathbf{v_1}^*, \mathbf{v_2}^*) = \arg\min_{\mathbf{v_1}, \mathbf{v_2}} I(\mathbf{v_1}; \mathbf{v_2})$, subject to $I(\mathbf{v_1}; \mathbf{y}) = I(\mathbf{v_2}; \mathbf{y}) = I(\mathbf{x}; \mathbf{y})$. Given $\mathbf{v_1}^*, \mathbf{v_2}^*$, the representation $\mathbf{z_1^*}$ (or $\mathbf{z_2^*}$) learned by contrastive learning is optimal for $\mathcal{T}$ (Def 3), thanks to the minimality and sufficiency of $f_1$ and $f_2$.*

Unlike in information bottleneck, for contrastive learning we often do not have access to a fully-labeled training set that specifies the downstream task in advance, and thus evaluating how much task-relevant information is contained in the views and representation at training time is challenging. Instead, the construction of views has typically been guided by domain knowledge that alters the input while preserving the task-relevant variable.

### 3.3 View Selection Influences Mutual Information and Accuracy

The above analysis suggests that transfer performance will be upper-bounded by a reverse-U shaped curve (Fig. 1b, right), with the sweet spot at the top of the curve. In theory, when the mutual information between views is changed, information about the downstream task and nuisance variables can be selectively included or excluded, biasing the learned representation, as shown in Fig. 2. The upper-bound reverse-U might not be reached if views are selected that share noise rather than signal. But practically, a recent study [53] suggests that the reverse-U shape is quite common. Here we show

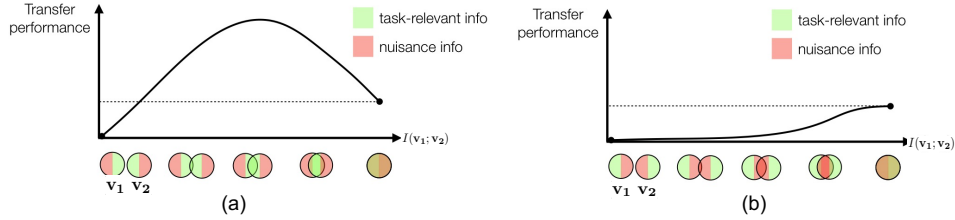

Figure 2: As the mutual information between views is changed, information about the downstream task (green) and nuisance variables (red) can be selectively included or excluded, biasing the learned representation. **(a)** depicts a scenario where views are chosen to preserve downstream task information between views while throwing out nuisance information, while in **(b)** reducing MI always throws out information relevant for the task leading to decreasing performance as MI is reduced.

several examples where reducing $I(\mathbf{v_1}; \mathbf{v_2})$ improves downstream accuracy. We use $I_{\text{NCE}}$ as a neural proxy for $I$, and note it depends on network architectures. Therefore for each plot in this paper, we only vary the input views while keeping other settings the same, to make the results comparable.

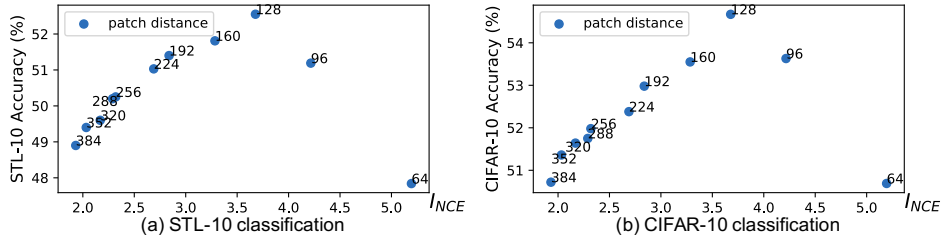

Figure 3: We create views by using pairs of image patches at various offsets from each other. As $I_{NCE}$ is reduced, the downstream task accuracy firstly increases and then decreases, leading to a reverse-U shape.

**Example 1:** Reducing $I(\mathbf{v_1}; \mathbf{v_2})$ with spatial distance. We create views by randomly cropping two patches of size 64x64 from the same image with various offsets. Namely, one patch starts at position $(x, y)$ while the other starts at $(x + d, y + d)$, with $(x, y)$ randomly generated. We increase $d$ from 64 to 384, and sample patches from inside high resolution images in the DIV2K dataset [1]. After contrastive training stage, we evaluate on STL-10 and CIFAR-10 by freezing the encoder and training a linear classifier. The plots in Fig. 3 shows the *Mutual Information* v.s. *Accuracy*. The results show that the reverse-U curve is consistent across both STL-10 and CIFAR-10. We can identify the sweet spot at $d = 128$. More details are provided in Appendix.

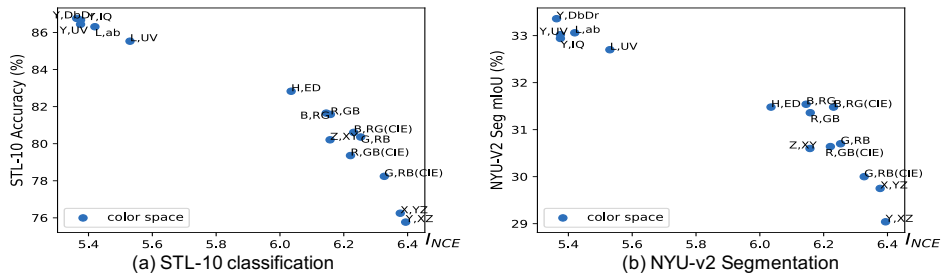

Figure 4: We build views by splitting channels of different color spaces. As $I_{\text{NCE}}$ decreases, the accuracy on downstream tasks (STL-10 classification, NYU-v2 segmentation) improves.

**Example 2:** Reducing $I(\mathbf{v_1}; \mathbf{v_2})$ with different color spaces. The correlation between channels may vary significantly across different color spaces. We follow [53, 64] to split each color space into two views, such as $\{Y, DbDr\}$ and $\{R, GB\}$. We perform contrastive learning on STL-10, and measure the representation quality by linear classification accuracy on the STL-10 and segmentation performance on NYU-V2 [40] images. As shown in Fig. 4, the downstream performance keeps increasing as $I_{\text{NCE}}$ decreases for both classification and segmentation. Here we do not observe the the left half of the reverse U-shape, but in Sec. 4.2 we will show a learning method that generates color spaces which reveal the full shape and touch the sweet spot.

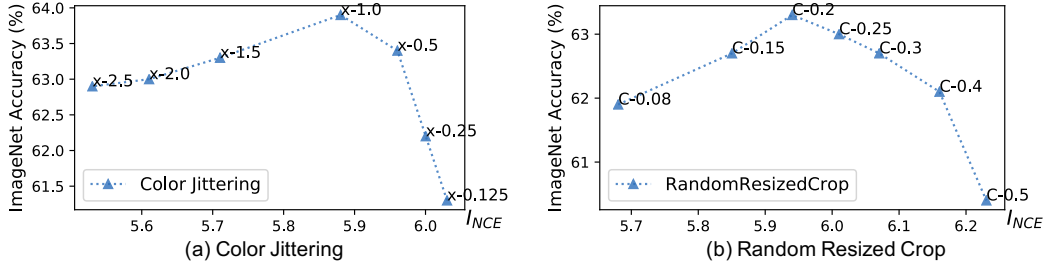

Figure 5: The reverse U-shape traced out by parameters of individual augmentation functions.

## 3.4 Data Augmentation to Reduce Mutual Information between Views

Multiple views can also be generated through augmenting an input in different ways. We can unify several recent contrastive learning methods through the perspective of view generation: despite differences in architecture, objective, and engineering tricks, all recent contrastive learning methods create two views $\mathbf{v_1}$ and $\mathbf{v_2}$ that implicitly follow the InfoMin principle. Below, we consider several recent works in this framework:

**InstDis [58] and MoCo [23].** These two methods create views by applying a stochastic data augmentation function twice to the same input: (1) sample an image $X$ from the empirical distribution $p(x)$; (2) sample two independent transformations $t_1, t_2$ from a distribution of data augmentation functions $\mathbb{T}$; (3) let $\mathbf{v}_1 = t_1(X)$ and $\mathbf{v}_2 = t_2(X)$.

**CMC [53].** CMC further split images across color channels such that $\mathbf{v}_1^{\text{cmc}}$ is the first color channel of $\mathbf{v}_1$, and $\mathbf{v}_2^{\text{cmc}}$ is the last two channels of $\mathbf{v}_2$. By this design, $I(\mathbf{v}_1^{\text{cmc}}; \mathbf{v}_2^{\text{cmc}}) \leq I(\mathbf{v}_1; \mathbf{v}_2)$ is theoretically guaranteed, and we observe that CMC performs better than InstDis.

**PIRL [38].** PIRL keeps $\mathbf{v}_1^{\text{pirl}} = \mathbf{v}_1$ but transforms the other view $\mathbf{v}_2$ with random JigSaw shuffling $h$ to get $\mathbf{v}_2^{\text{pirl}} = h(\mathbf{v}_2)$. Similary we have $I(\mathbf{v}_1^{\text{pirl}}; \mathbf{v}_2^{\text{pirl}}) \leq I(\mathbf{v}_1; \mathbf{v}_2)$ as $h(\cdot)$ introduces randomness.

**SimCLR [7].** Despite other engineering techniques and tricks, SimCLR uses a stronger class of augmentations $\mathbb{T}'$, which leads to smaller mutual information between the two views than InstDis.

**CPC [42].** Different from the above methods that create views at the image level, CPC gets views $\mathbf{v}_1^{\text{cpc}}, \mathbf{v}_2^{\text{cpc}}$ from local patches with strong data augmentation (e.g., RA [11]) which results in smaller $I(\mathbf{v}_1^{\text{cpc}}; \mathbf{v}_2^{\text{cpc}})$. As in Sec. 3.3, cropping views from disjoint patches also reduces $I(\mathbf{v}_1^{\text{cpc}}; \mathbf{v}_2^{\text{cpc}})$.

Table 1: Single-crop ImageNet accuracies (%) of linear classifiers [63] trained on representations learned with different contrastive methods using ResNet-50 [24]. InfoMin Aug. refers to data augmentation using *RandomResizedCrop*, *Color Jittering*, *Gaussian Blur*, *RandAugment*, *Color Dropping*, and a *JigSaw* branch as in PIRL [38]. * indicates splitting the network into two halves.

| Method | Architecture | Param. | Head | Epochs | Top-1 | Top-5 |
|---|---|---|---|---|---|---|
| InstDis [58] | ResNet-50 | 24 | Linear | 200 | 56.5 | - |
| Local Agg. [67] | ResNet-50 | 24 | Linear | 200 | 58.8 | - |
| CMC [53] | ResNet-50* | 12 | Linear | 240 | 60.0 | 82.3 |
| MoCo [23] | ResNet-50 | 24 | Linear | 200 | 60.6 | - |
| PIRL [38] | ResNet-50 | 24 | Linear | 800 | 63.6 | - |
| CPC v2 [25] | ResNet-50 | 24 | - | - | 63.8 | 85.3 |
| SimCLR [7] | ResNet-50 | 24 | MLP | 1000 | 69.3 | 89.0 |
| InfoMin Aug. (Ours) | ResNet-50 | 24 | MLP | 200 | 70.1 | 89.4 |
| InfoMin Aug. (Ours) | ResNet-50 | 24 | MLP | 800 | **73.0** | **91.1** |

Besides, we also analyze how changing the magnitude parameter of individual augmentation functions trances out reverse-U shapes. We consider *RandomResizedCrop* and *Color Jittering*. For the former, a parameter c sets a low-area cropping bound, and smaller c indicates stronger augmentation. For the latter, a parameter x is adopted to control the strengths. The plots on ImageNet [12] are shown in Fig. 5, where we identify a sweet spot at $1.0$ for *Color Jittering* and $0.2$ for *RandomResizedCrop*.

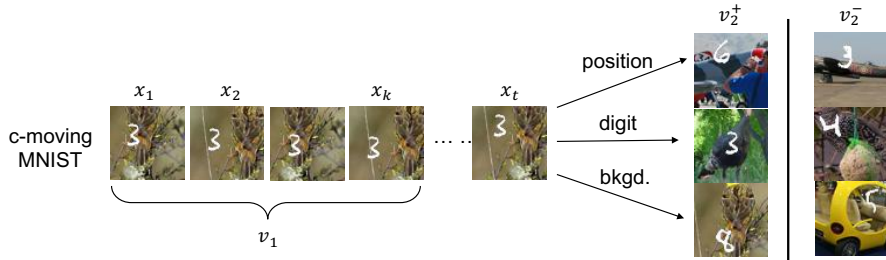

Figure 6: Illustration of the Colorful-Moving-MNIST dataset. In this example, the first view $\mathbf{v_1}$ is a sequence of frames containing the moving digit, e.g., $\mathbf{v_1} = x_{1:k}$. The matched second view $\mathbf{v_2^+}$ share some factor with $x_t$ that $\mathbf{v_1}$ can predict, while the unmatched view $\mathbf{v_2^-}$ does not share factor with $x_t$.

Table 2: We study how information shared by views $I(\mathbf{v_1}; \mathbf{v_2})$ would affect the representation quality, by evaluating on three downstream tasks: digit classification, localization, and background (STL-10) classification. Evaluation for contrastive methods is performed by freezing the backbone and training a linear task-specific head

|  | $I(\mathbf{v_1}; \mathbf{v_2})$ | digit cls. error rate (%) | background cls. error rate (%) | digit loc. error pixels |
|---|---|---|---|---|
| Single Factor | *digit* | **16.8** | 88.6 | 13.6 |
| | *bkgd* | 88.6 | **51.7** | 16.1 |
| | *pos* | 57.9 | 87.6 | **3.95** |
| Multiple Factors | *bkgd, digit, pos* | 88.8 | 56.3 | 16.2 |
| | *bkgd, digit* | 88.2 | 53.9 | 16.3 |
| | *bkgd, pos* | 88.8 | 53.8 | 15.9 |
| | *digit, pos* | **14.5** | 88.9 | 13.7 |
| Supervised | | 3.4 | 45.3 | 0.93 |

Motivated by the InfoMin principle, we propose a new set of data augmentation, called *InfoMin Aug*. In combination of the JigSaw strategy proposed in PIRL [38], our InfoMin Aug achieves 73.0% top-1 accuracy on ImageNet linear readout benchmark with ResNet-50, outperforming SimCLR [7] by nearly 4%, as shown in Table 1. Besides, we also found that transferring our unsupervisedly pre-trained models to PASCAL VOC object detection and COCO instance segmentation consistently outperforms supervised ImageNet pre-training. More details and results are in Appendix.

# 4 Learning views for contrastive learning

Hand-designed data augmentation is an effective method for generating views that have reduced mutual information and strong transfer performance for images. However, as contrastive learning is applied to new domains, generating views through careful construction of data augmentation strategies may prove ineffective. Furthermore, the types of views that are useful depend on the downstream task. Here we show the task-dependence of optimal views on a simple toy problem, and propose an unsupervised and semi-supervised learning method to *learn* views from data.

## 4.1 Optimal Views Depend on the Downstream Task

To understand how the choice of views impact the representations learned by contrastive learning, we construct a toy dataset that mixes three tasks. We build our toy dataset by combining **Moving-MNIST** [51] (consisting of videos where digits move inside a black canvas with constant speed and bounce off of image boundaries), with a fixed background image sampled from the STL-10 dataset [10]. We call this dataset **Colorful Moving-MNIST**, which consists of three factors of variation in each frame: *the class of the digit*, *the position of the digit*, and *the class of background image* (see Appendix for more details). Here we analyze how the choice of views impacts which of these factors are extracted by contrastive learning.

**Setup.** We fix view $\mathbf{v_1}$ as the sequence of past frames $\mathbf{x}_{1:k}$. For simplicity, we consider $\mathbf{v_2}$ as a single image, and construct it by referring to frame $\mathbf{x}_{t(t>k)}$. One example of visualization is shown in Fig. 6, and please refer to Appendix for more details. We consider 3 downstream tasks for an image: (1) predict the digit class; (2) localize the digit; (3) classify the background image (10 classes from STL-10). This is performed by freezing the backbone and training a linear task-specific head. We also provide a "supervised" baseline that is trained end-to-end for comparison.

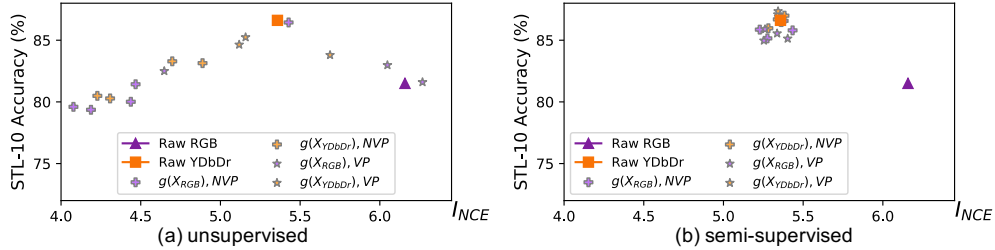

Figure 7: View generator learned by (a) unsupervised or (b) semi-supervised objectives.

**Single Factor Shared.** We consider the case that $\mathbf{v_1}$ and $\mathbf{v_2}$ only share one of the three factors: *digit*, *position*, or *background*. We synthesize $\mathbf{v_2}$ by setting one of the three factors the same as $\mathbf{x}_t$ but randomly picking the other two. In such cases, the mutual information $I(\mathbf{v_1}; \mathbf{v_2})$ is either about *digit*, *position*, or *background*. The results are summarized in Table 2, which clearly shows that the performance is significantly affected by what is shared between $\mathbf{v_1}$ and $\mathbf{v_2}$. Specifically, if the downstream task is relevant to one factor, $I(\mathbf{v_1}; \mathbf{v_2})$ should include that factor rather than others. For example, when $\mathbf{v_2}$ only shares background image with $\mathbf{v_1}$, contrastive learning can hardly learn representations that capture digit class and location.

**Multiple Factors Shared.** We further explore how representation quality is changed if $\mathbf{v_1}$ and $\mathbf{v_2}$ share multiple factors. We follow a similar procedure as above to control factors shared by $\mathbf{v_1}$ and $\mathbf{v_2}$, and present the results in Table 2. We found that one factor can overwhelm another; for instance, whenever *background* is shared, the latent representation leaves out information for discriminating or localizing digits. This might because the information bits of background predominates, and the encoder chooses the background as a "shortcut" to solve the contrastive pre-training task. When $\mathbf{v_1}$ and $\mathbf{v_2}$ share *digit* and *position*, the former is preferred over the latter.

### 4.2 Synthesizing Views with Invertible Generators

In this section, we design unsupervised and semi-supervised methods that synthesize novel views following the InfoMin principle. Concretely, we extend the color space experiments in Sec. 3.3 by learning flow-based models [14, 13, 31] that transfer natural color spaces into novel color spaces, from which we split the channels to get views. We still call the output of flow-based models as color spaces because the flows are designed to be pixel-wise and bijective (by its nature), which follows the property of color space conversion. After the views have been learned, we perform standard contrastive learning followed by linear classifier evaluation.

Practically, the flow-based model $g$ is restricted to pixel-wise 1x1 convolutions and ReLU activations, operating independently on each pixel. We try both volume preserving (VP) and non-volume preserving (NVP) flows. For an input image $X$, the splitting over channels is represented as $\{X_1, X_{2:3}\}$. $\hat{X}$ signifies the transformed image, i.e., $\hat{X} = g(X)$. Experiments are conducted on STL-10, which includes 100k unlabeled and 5k labeled images. More details are in Appendix.

#### 4.2.1 Unsupervised View Learning: Minimize $I(\mathbf{v_1}; \mathbf{v_2})$

The idea is to leverage an adversarial training strategy [20]. Given $\hat{X} = g(X)$, we train two encoders $f_1, f_2$ to maximize $I_{\text{NCE}}(\hat{X}_1; \hat{X}_{2:3})$ as in Eqn. 1, similar to the discriminator of GAN [20]. Meanwhile, $g$ is adversarially trained to minimize $I_{\text{NCE}}(\hat{X}_1; \hat{X}_{2:3})$. Formally, the objective is:

$$\min_{g} \max_{f_1, f_2} I_{\text{NCE}}^{f_1, f_2}\big(g(X)_1; g(X)_{2:3}\big) \tag{2}$$

Alternatively, one may use other MI bounds [5, 45], but we find $I_{\text{NCE}}$ works well and keep using it. We note that the invertibility of $g(\cdot)$ prevent it from learning degenerate/trivial solutions.

**Results.** We experiment with *RGB* and *YDbDr*. As shown in Fig. 7(a), a reverse U-shape of $I_{\text{NCE}}$ and downstream accuracy is present. Interestingly, YDbDr is already near the sweet spot. This happens to be in line with our human prior that the "luminance-chrominance" decomposition is a

Table 3: Comparison of different view generators by measuring STL-10 classification accuracy: *supervised*, *unsupervised*, and *semi-supervised*. "# of Images" indicates how many images are used to learn view generators. In representation learning stage, all 105k images are used.

| Method (# of Images) | RGB | YDbDr |
|---|---|---|
| unsupervised (100k) | $82.4 \pm 3.2$ | $84.3 \pm 0.5$ |
| supervised (5k) | $79.9 \pm 1.5$ | $78.5 \pm 2.3$ |
| semi-supervised (105k) | $\mathbf{86.0 \pm 0.6}$ | $\mathbf{87.0 \pm 0.3}$ |
| raw views | $81.5 \pm 0.2$ | $86.6 \pm 0.2$ |

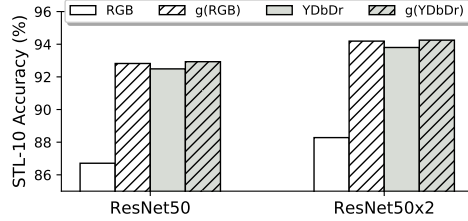

Figure 8: Switching to larger backbones with views learned by the semi-supervised method.

good way to decorrelate colors but still retains recognizability of objects. We also note that another luminance-chrominance decomposition Lab, which performs similarly well to YDbDr (Fig. 4), was designed to mimic the way humans perceive color [28]. Our analysis therefore suggests yet another rational explanation for why humans perceive color the way we do – human perception of color may be near optimal for self-supervised representation learning.

With this unsupervised objective, in most cases $I_{\text{NCE}}$ between views is overly reduced. In addition, we found this GAN-style training is unstable, as different runs with the same hyper-parameter vary significantly. We conjecture it is because the view generator has no knowledge about the downstream task, and thus the constraint $I(\mathbf{v_1}, \mathbf{y}) = I(\mathbf{v_2}, \mathbf{y}) = I(\mathbf{x}, \mathbf{y})$ in Proposition 3.1 is heavily broken. To overcome this, we further develop an semi-supervised view learning method.

### 4.2.2 Semi-supervised View Learning: Find Views that Share the Label Information

We assume a handful of labels for the downstream task are available. Thus we can guide the generator $g$ to retain $I(g(X)_1, \mathbf{y})$ and $I(g(X)_{2:3}, \mathbf{y})$. Practically, we introduce two classifiers on each of the learned views to perform classification during the view learning process. Formally, we optimize:

$$\min_{g,c_1,c_2} \max_{f_1,f_2} \underbrace{I_{\text{NCE}}^{f_1,f_2}(g(X)_1; g(X)_{2:3})}_{\text{unsupervised: reduce } I(v_1;v_2)} + \underbrace{\mathcal{L}_{ce}(c_1(g(X)_1), y) + \mathcal{L}_{ce}(c_2(g(X)_{2:3}), y)}_{\text{supervised: keep } I(v_1;y) \text{ and } I(v_2;y)} \quad (3)$$

where $c_1, c_2$ are the classifiers. The $I_{\text{NCE}}$ term applies to all data while the latter two are only for labeled data. In each iteration, we sample an unlabeled batch and a labeled batch. After this process is done, we use frozen $g$ to generate views for *unsupervised* contrastive representation learning.

**Results.** The plots are shown in Figure 7(b). Now the learned views are centered around the sweet spot, no matter what the input color space is and whether the generator is VP or NVP, which highlights the importance of keeping information about $\mathbf{y}$. Meanwhile, to see the importance of the unsupervised term, which reduces $I_{NCE}$, we train another view generator with only supervised loss. We further compare "supervised", "unsupervised" and "semi-supervised" (the supervised + unsupervised losses) generators in Table 3, where we also includes contrastive learning over the original color space ("raw views") as a baseline. The semi-supervised view generator significantly outperforms the supervised one, validating the importance of reducing $I(\mathbf{v_1}; \mathbf{v_2})$. We compare further compare $g(X)$ with $X$ ($X$ is RGB or YDbDr) on larger backbone networks, as shown in Fig. 8, We see that the learned views consistently outperform its raw input, e.g., $g(RGB)$ surpasses $RGB$ by a large margin and reaches $94\%$ classification accuracy.

## 5 Conclusion

We have characterized that good views for a given task in contrastive representation learning framework should retain task-relevant information while minimizing irrelevant nuisances, which we call *InfoMin* principle. Based on it, we demonstrate that optimal views are task-dependent in both theory and practice. We further propose a semi-supversied method to learn effective views for a given task. In addition, we analyze the data augmentation used in recent methods from the *InfoMin* perspective, and further propose a new set of data augmentation that achieved a new state-of-the-art top-1 accuracy on ImageNet linear readout benchmark with a ResNet-50.

## Broader Impact

This paper is on the basic science of representation learning, and we believe it will be beneficial to both the theory and practice of this field. An immediate application of self-supervised representation learning is to reduce the reliance on labeled data for downstream applications. This may have the beneficial effects of being more cost effective and reducing biases introduced by human annotations. At the same time, these methods open up the ability to use uncurated data more effectively, and such data may hide errors and biases that would have been uncovered via the human curation process. We also note that the view constructions we propose are not bias free, even when they do not use labels: using one color space or another may hide or reveal different properties of the data. The choice of views therefore plays a similar role to the choice of training data and training annotations in traditional supervised learning.

## Acknowledgments and Disclosure of Funding

**Acknowledgements.** This work was done when Yonglong Tian was a student researcher at Google. We thank Kevin Murphy for fruitful and insightful discussion; Lucas Beyer for feedback on related work; and Google Cloud team for supporting computation resources. Yonglong is grateful to Zhoutong Zhang for encouragement and feedback on experimental design.

**Funding.** Funding for this project was provided Google, as part of Yonglong Tian's role as a student researcher at Google.

**Competing interests.** In the past 36 months, Phillip Isola has had employment at MIT, Google, and OpenAI; honorarium for lecturing at the ACDL summer school in Italy; honorarium for speaking at GIST AI Day in South Korea. P.I.'s lab at MIT has been supported by grants from Facebook, IBM, and the US Air Force; start up funding from iFlyTech via MIT; gifts from Adobe and Google; compute credit donations from Google Cloud. Yonglong Tian is a Ph.D. student supported by MIT EECS department. Chen Sun, Ben Poole, Dilip Krishan, and Cordelia Schmid are employees at Google.

## Footnotes

[1]Project page: http://hobbitlong.github.io/InfoMin

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
