[Supplementary Material]

# Supplementary: What Makes for Good Views for Contrastive Learning?

**Yonglong Tian***
MIT CSAIL

**Chen Sun**
Google, Brown University

**Ben Poole**
Google Research

**Dilip Krishnan**
Google Research

**Cordelia Schmid**
Google Research

**Phillip Isola**
MIT CSAIL

## A  Proof of Proposition 3.1

In this section, we provide proof for the statement regarding optimal views in proposition 3.1 of the main text. As a warmup, we firstly recap some properties of mutual information.

### A.1  Properties of MI [5]:

(1) Nonnegativity:

$$I(\mathbf{x}; \mathbf{y}) \geq 0; I(\mathbf{x}; \mathbf{y}|\mathbf{z}) \geq 0$$

(2) Chain Rule:

$$I(\mathbf{x}; \mathbf{y}, \mathbf{z}) = I(\mathbf{x}; \mathbf{y}) + I(\mathbf{x}; \mathbf{z}|\mathbf{y})$$

(2) Multivariate Mutual Information:

$$I(\mathbf{x}_1; \mathbf{x}_2; ...; \mathbf{x}_{n+1}) = I(\mathbf{x}_1; ...; \mathbf{x}_n) - I(\mathbf{x}_1; ...; \mathbf{x}_n|\mathbf{x}_{n+1})$$

### A.2  Proof

**Proposition A.1.** *According to Proposition 1, the optimal views $\mathbf{v}_1^*, \mathbf{v}_2^*$ for task $\mathcal{T}$ with label $\mathbf{y}$, are views such that $I(\mathbf{v}_1^*; \mathbf{v}_2^*) = I(\mathbf{v}_1^*; \mathbf{y}) = I(\mathbf{v}_2^*; \mathbf{y}) = I(\mathbf{x}; \mathbf{y})$*

*Proof.* Since $I(\mathbf{v}_1; \mathbf{y}) = I(\mathbf{v}_2; \mathbf{y}) = I(\mathbf{x}; \mathbf{y})$, and $\mathbf{v}_1, \mathbf{v}_2$ are functions of $\mathbf{x}$.

$$\begin{aligned}
I(\mathbf{y}; \mathbf{x}) &= I(\mathbf{y}; \mathbf{v}_1, \mathbf{v}_2) \\
&= I(\mathbf{y}; \mathbf{v}_1) + I(\mathbf{y}; \mathbf{v}_2|\mathbf{v}_1) \\
&= I(\mathbf{y}; \mathbf{x}) + I(\mathbf{y}; \mathbf{v}_2|\mathbf{v}_1)
\end{aligned}$$

Therefore $I(\mathbf{y}; \mathbf{v}_2|\mathbf{v}_1) = 0$, due to the nonnegativity. Then we have:

$$\begin{aligned}
I(\mathbf{v}_1; \mathbf{v}_2) &= I(\mathbf{v}_1; \mathbf{v}_2) + I(\mathbf{y}; \mathbf{v}_2|\mathbf{v}_1) \\
&= I(\mathbf{v}_2; \mathbf{v}_1, \mathbf{y}) \\
&= I(\mathbf{v}_2; \mathbf{y}) + I(\mathbf{v}_2; \mathbf{v}_1|\mathbf{y}) \\
&\geq I(\mathbf{v}_2; \mathbf{y}) = I(\mathbf{x}; \mathbf{y})
\end{aligned}$$

Therefore the optimal views $\mathbf{v}_1^*, \mathbf{v}_2^*$ that minimizes $I(\mathbf{v}_1; \mathbf{v}_2)$ subject to the constraint yields $I(\mathbf{v}_1^*; \mathbf{v}_2^*) = I(\mathbf{x}; \mathbf{y})$. Also note that optimal views $\mathbf{v}_1^*, \mathbf{v}_2^*$ are conditionally independent given $\mathbf{y}$, as now $I(\mathbf{v}_2^*; \mathbf{v}_1^*|\mathbf{y}) = 0$. $\qquad\square$

**Proposition A.2.** *Given optimal views $\mathbf{v}_1^*, \mathbf{v}_2^*$ and minimal sufficient encoders $f_1$, $f_2$, then the learned representations $\mathbf{z}_1$ (or $\mathbf{z}_2$) are sufficient statistic of $\mathbf{v}_1$ (or $\mathbf{v}_2$) for $\mathbf{y}$, i.e., $I(\mathbf{z}_1; \mathbf{y}) = I(\mathbf{v}_1; \mathbf{y})$ or $I(\mathbf{z}_2; \mathbf{y}) = I(\mathbf{v}_2; \mathbf{y})$.*

*Proof.* Let's prove for $\mathbf{z}_1$. Since $\mathbf{z}_1$ is a function of $\mathbf{v}_1$, we have:

$$\begin{aligned}
I(\mathbf{y}; \mathbf{v}_1) &= I(\mathbf{y}; \mathbf{v}_1, \mathbf{z}_1) \\
&= I(\mathbf{y}; \mathbf{z}_1) + I(\mathbf{y}; \mathbf{v}_1 | \mathbf{z}_1)
\end{aligned}$$

To prove $I(\mathbf{y}; \mathbf{v}_1) = I(\mathbf{y}; \mathbf{z}_1)$, we need to prove $I(\mathbf{y}; \mathbf{v}_1 | \mathbf{z}_1) = 0$.

$$\begin{aligned}
I(\mathbf{y}; \mathbf{v}_1 | \mathbf{z}_1) &= I(\mathbf{y}; \mathbf{v}_1) - I(\mathbf{y}; \mathbf{v}_1; \mathbf{z}_1) \\
&= I(\mathbf{y}; \mathbf{v}_1; \mathbf{v}_2) + I(\mathbf{y}; \mathbf{v}_1 | \mathbf{v}_2) - I(\mathbf{y}; \mathbf{v}_1; \mathbf{z}_1) \\
&= I(\mathbf{y}; \mathbf{v}_1; \mathbf{v}_2) + I(\mathbf{y}; \mathbf{v}_1 | \mathbf{v}_2) - [I(\mathbf{y}; \mathbf{v}_1; \mathbf{z}_1; \mathbf{v}_2) + I(\mathbf{y}; \mathbf{v}_1; \mathbf{z}_1 | \mathbf{v}_2)] \\
&= I(\mathbf{y}; \mathbf{v}_1 | \mathbf{v}_2) + [I(\mathbf{y}; \mathbf{v}_1; \mathbf{v}_2) - I(\mathbf{y}; \mathbf{v}_1; \mathbf{z}_1; \mathbf{v}_2)] - I(\mathbf{y}; \mathbf{v}_1; \mathbf{z}_1 | \mathbf{v}_2) \\
&= I(\mathbf{y}; \mathbf{v}_1 | \mathbf{v}_2) + I(\mathbf{y}; \mathbf{v}_1; \mathbf{v}_2 | \mathbf{z}_1) - I(\mathbf{y}; \mathbf{v}_1; \mathbf{z}_1 | \mathbf{v}_2) \\
&= I(\mathbf{y}; \mathbf{v}_1 | \mathbf{v}_2) + I(\mathbf{y}; \mathbf{v}_1; \mathbf{v}_2 | \mathbf{z}_1) + \underbrace{I(\mathbf{y}; \mathbf{z}_1 | \mathbf{v}_1, \mathbf{v}_2)}_{0} - I(\mathbf{y}; \mathbf{z}_1 | \mathbf{v}_2) \\
&\leq I(\mathbf{y}; \mathbf{v}_1 | \mathbf{v}_2) + I(\mathbf{y}; \mathbf{v}_1; \mathbf{v}_2 | \mathbf{z}_1) \\
&= I(\mathbf{y}; \mathbf{v}_1 | \mathbf{v}_2) + I(\mathbf{v}_1; \mathbf{v}_2 | \mathbf{z}_1) - \underbrace{I(\mathbf{v}_1; \mathbf{v}_2 | \mathbf{y}, \mathbf{z}_1)}_{0} \\
&= I(\mathbf{y}; \mathbf{v}_1 | \mathbf{v}_2) + I(\mathbf{v}_1; \mathbf{v}_2 | \mathbf{z}_1)
\end{aligned}$$

In the above derivation $I(\mathbf{y}; \mathbf{z}_1 | \mathbf{v}_1, \mathbf{v}_2) = 0$ because $\mathbf{z}_1$ is a function of $\mathbf{v}_1$; $I(\mathbf{v}_1; \mathbf{v}_2 | \mathbf{y}, \mathbf{z}_1) = 0$ because optimal views $\mathbf{v}_1, \mathbf{v}_2$ are conditional independent given $\mathbf{y}$, see Proposition A.1. Now, we can easily prove $I(\mathbf{y}; \mathbf{v}_1 | \mathbf{v}_2) = 0$ following a similar procedure in Proposition A.1. If we can further prove $I(\mathbf{v}_1; \mathbf{v}_2 | \mathbf{z}_1) = 0$, then we get $I(\mathbf{y}; \mathbf{v}_1 | \mathbf{z}_1) \leq 0$. By nonnegativity, we will have $I(\mathbf{y}; \mathbf{v}_1 | \mathbf{z}_1) = 0$.

To see $I(\mathbf{v}_1; \mathbf{v}_2 | \mathbf{z}_1) = 0$, recall that our encoders are sufficient. According to Definition 1, we have $I(\mathbf{v}_1; \mathbf{v}_2) = I(\mathbf{v}_2; \mathbf{z}_1)$:

$$\begin{aligned}
I(\mathbf{v}_1; \mathbf{v}_2 | \mathbf{z}_1) &= I(\mathbf{v}_1; \mathbf{v}_2) - I(\mathbf{v}_1; \mathbf{v}_2; \mathbf{z}_1) \\
&= I(\mathbf{v}_1; \mathbf{v}_2) - I(\mathbf{v}_2; \mathbf{z}_1) + \underbrace{I(\mathbf{v}_2; \mathbf{z}_1 | \mathbf{v}_1)}_{0} \\
&= 0
\end{aligned}$$

$\square$

**Proposition A.3.** *The representations $z_1$ and $z_2$ are also minimal for $y$.*

*Proof.* For all sufficient encoders, we have proved $\mathbf{z}_1$ are sufficient statistic of $\mathbf{v}_1$ for predicting $\mathbf{y}$. Namely $I(\mathbf{v}_1; \mathbf{y} | \mathbf{z}_1) = 0$. Now:

$$\begin{aligned}
I(\mathbf{z}_1; \mathbf{v}_1) &= I(\mathbf{z}_1; \mathbf{v}_1 | \mathbf{y}) + I(\mathbf{z}_1; \mathbf{v}_1; \mathbf{y}) \\
&= I(\mathbf{z}_1; \mathbf{v}_1 | \mathbf{y}) + I(\mathbf{v}_1; \mathbf{y}) - \underbrace{I(\mathbf{v}_1; \mathbf{y} | \mathbf{z}_1)}_{0} \\
&= I(\mathbf{z}_1; \mathbf{v}_1 | \mathbf{y}) + I(\mathbf{v}_1; \mathbf{y}) \\
&\geq I(\mathbf{v}_1; \mathbf{y})
\end{aligned}$$

The minimal sufficient encoder will minimize $I(\mathbf{z}_1; \mathbf{v}_1)$ to $I(\mathbf{v}_1; \mathbf{y})$. This is achievable and leads to $I(\mathbf{z}_1; \mathbf{v}_1 | \mathbf{y}) = 0$. Therefore, $z_1$ is a minimal sufficient statistic for predicting $y$, thus optimal. Similarly, $z_2$ is also optimal.

$\square$

# B Implementation Details

## B.1 Spatial Patches with Different Distance

**Why using DIV2K [1]?** Recall that we randomly sample patches with a distance of $d$. During such sampling process, there is a possible bias that with an image of relatively small size (e.g., 512x512), a large $d$ (e.g., 384) will always push these two patches around the boundary. To minimize this bias, we choose to use high resolution images (e.g. 2k) from DIV2K dataset.

**Setup and Training.** We use the training framework of CMC [28]. The backbone network is a tiny AlexNet, following [17, 28]. We train for 3000 epochs, with the learning rate initialized as 0.03 and decayed with cosine annealing.

**Evaluation.** We evaluate the learned representation on both STL-10 and CIFAR-10 datasets. For CIFAR-10, we resize the image to $64\times64$ to extract features. The linear classifier is trained for 100 epochs.

## B.2 Channel Splitting with Various Color Spaces

**Setup and Training.** The backbone network is also a tiny AlexNet, with the modification of adapting the first layer to input of 1 or 2 channels. We follow the training recipe in [28].

**Evaluation.** For the evaluation on STL-10 dataset, we train a linear classifier for 100 epochs and report the single-crop classification accuracy. For NYU-Depth-v2 segmentation task, we freeze the backbone network and train a 4-layer decoder on top of the learned representations. We report the mean IoU for labeled classes.

## B.3 Reducing $I(\mathbf{v_1}; \mathbf{v_2})$ with Frequency Separation

(a) STL-10 classification    (b) Tiny ImageNet classification

Figure 1: We create views by splitting images into low- and high-frequency pairs with a blur function parameterized by $\sigma$. $I_{NCE}$ is maximized at $\sigma = 0.7$. Starting from this point, either increasing or decreasing $\sigma$ will reduce $I_{NCE}$ but interestingly they form two different trajectories. When increasing $\sigma$ from 0.7, the accuracy firstly improves and then drops, forming a reverse-U shape corresponding to (a) in Figure 2 of the main paper. While decreasing $\sigma$ from 0.7, the accuracy keeps diminishing, corresponding to (b) in Figure 2 of the main paper.

Another example we consider is to separate images into low- and high-frequency images. To simplify, we extract $\mathbf{v_1}$ and $\mathbf{v_2}$ by Gaussian blur, i.e.,

$$\mathbf{v_1} = \texttt{Blur}(\mathbf{x}, \sigma)$$
$$\mathbf{v_2} = \mathbf{x} - \mathbf{v_1}$$

where $\texttt{Blur}$ is the Gaussian blur function and $\sigma$ is the parameter controlling the kernel. Extremely small or large $\sigma$ can make the high- or low-frequency image contain little information. In theory, the maximal $I(\mathbf{v_1}; \mathbf{v_2})$ is obtained with some intermediate $\sigma$. As shown in Figure 1, we found $\sigma = 0.7$ leads to the maximal $I_{NCE}$ on the STL-10 dataset. Either blurring more or less will reduce $I_{NCE}$, but interestingly blurring more leads to different trajectory in the plot than blurring less. When increasing $\sigma$ from 0.7, the accuracy firstly improves and then drops, forming a reverse-U shape with a sweet spot at $\sigma = 1.0$. This situation corresponds to (a) in Figure 2 of the main paper. While decreasing $\sigma$

Figure 2: Illustration of the Colorful-Moving-MNIST dataset. In this example, the first view $\mathbf{v_1}$ is a sequence of frames containing the moving digit, e.g., $\mathbf{v_1} = x_{1:k}$. The matched second view $\mathbf{v_2^+}$ share some factor with $x_t$ that $\mathbf{v_1}$ can predict, while the unmatched view $\mathbf{v_2^-}$ does not share factor with $x_t$.

from 0.7, the accuracy keeps diminishing, corresponding to (b) in Figure 2 of the main paper. This reminds us of the two aspects in Proposition 3.1: mutual information is not the whole story; *what* information is shared between the two views also matters.

**Setup and Training.** The setup is almost the same as that in color channel splitting experiments, except that each view consists of three input channels. We follow the training recipe in [28].

**Evaluation.** We train a linear classifier for 100 epochs on STL-10 dataset and 40 epochs on TinyImageNet dataset.

### B.4  Colorful Moving MNIST

**Dataset.** Following the original Moving MNIST dataset [27], we use a canvas of size $64\times64$, which contains a digit of size $28\times28$. The back ground image is a random crop from original STL-10 images ($96\times96$). The starting position of the digit is uniformly sampled inside the canvas. The direction of the moving velocity is uniformly sampled in $[0, 2\pi]$, while the magnitude is kept as $0.1$ of the canvas size. When the digit touches the boundary, the velocity is reflected.

**Setup.** We use the first 10 frames as $\mathbf{v_1}$ (namely $k = 10$), and we construct $\mathbf{v_2}$ by referring to the 20-th frame (namely $t = 20$). During the contrastive learning phase, we employ a 4-layer ConvNet to encode images and use a single layer LSTM [18] on top of the ConvNet to aggregate features of continuous frames. The CNN backbone consists of 4 layers with $8, 16, 32, 64$ filters from low to high. Average pooling is applied after the last convolutional layer, resulting in a 64 dimensional representation. The dimensions of the hidden layer and output in LSTM are both 64.

**Examples**. The examples of $\mathbf{v_1}$ and $\mathbf{v_2}$ are shown in Figure 2, where the three rows on the RHS shows cases that only a single factor (digit, position, or background) is shared.

**Training.** We perform intra-batch contrast. Namely, inside each batch of size 128, we contrast each sample with the other 127 samples. We train for 200 epochs, with the learning rate initialized as $0.03$ and decayed with cosine annealing.

### B.5  Un-/Semi-supervised View Learning

Figure 3: Volume-preserving (a), and none volume-preserving (b) invertible model.

**Invertible Generator.** Figure 3 shows the basic building block for the Volume-Preserving (VP) and None-Volume-Preserving (NVP) invertible view generator. The $F$ and $G$ are pixel-wise convolutional function, *i.e.*, convolutional layers with $1\times1$ kernel. $\mathbf{X}_1$ and $\mathbf{Y}_1$ represent a single channel of the input and output respectively, while $\mathbf{X}_2$ and $\mathbf{Y}_2$ represent the other two channels. While stacking

basic building blocks, we alternatively select the first, second, and the third channel as $\mathbf{X}_1$, to enhance the expressivity of view generator.

**Setup and Training.** For unsupervised view learning that only uses the adversarial $I_{NCE}$ loss, we found the training is relatively unstable, as also observed in GAN [12]. We found the learning rate of view generator should be larger than that of $I_{NCE}$ approximator. Concretely, we use Adam optimizer [19], and we set the learning rates of view generator and $I_{NCE}$ approximator as $2e$-$4$ and $6e$-$4$, respectively. For the semi-supervised view learning, we found the training is stable across different learning rate combinations, which we considered as an advantage. To be fair, we still use the same learning rates for both view generator and $I_{NCE}$ approximator.

**Contrastive Learning and Evaluation.** After the view learning stage, we perform contrastive learning and evaluation by following the recipe in Section B.2.

## C Data Augmentation as InfoMin

### C.1 InfoMin Augmentation

```
PyTorch-style data augmentation

RandomResizedCrop(scale=(0.2, 1.0))
RandomHorizontalFlip()
# CJ(x): random color jitter with x
cj = ColorJitter([0.8,0.8,0.8,0.4]*x)
RandomApply([cj], p=0.8)
# Blur: random blurring
blur = Blur(sigma=(0.1,2.0))
RandomApply([blur], p=0.5)
# RA: RandAugment
rnd_augment()
RandomGrayscale(p=0.2),
```

(a) $I_{NCE}$ $v.s$ Accuracy          (b) Data Augmentation

Figure 4: (a) data augmentation as InfoMin on ImageNet with linear projection head; (b) illustration of step-by-step data augmentation used in InfoMin.

**InfoMin Aug.** We gradually strengthen the family of data augmentation functions $\mathbb{T}$, and plot the trend between accuracy in downstream linear evaluation benchmarks and $I_{NCE}$. The overall results are shown in Figure 4(a), where the plot is generated by only varying data augmentation while keeping all other settings fixed. We consider *Color Jittering* with various strengths, *Gaussian Blur*, *RandAugment* [6], and their combinations, as illustrated in Figure 4(b). The results suggest that as we reduce $I_{NCE}(\mathbf{v_1}; \mathbf{v_1})$, via stronger $\mathbb{T}$ (in theory, $I(\mathbf{v_1}; \mathbf{v_1})$ also decreases), the downstream accuracy keeps improving.

### C.2 Analysis of Data Augmentation as it relates to MI and Transfer Performance

We also investigate how sliding the strength parameter of individual augmentation functions leads to a practical reverse-U curves, as shown in Figures 5 and 6.

**Cropping.** In PyTorch, the `RandomResizedCrop(scale=(c, 1.0))` data augmentation function sets a low-area cropping bound `c`. Smaller `c` means more aggressive data augmentation. We vary `c` for both a linear critic head [30] (with temperature 0.07) and nonlinear critic head [3] (with temperature 0.15), as shown in Figure 5. In both cases, decreasing `c` forms a reverse-U shape between $I_{NCE}$ and linear classification accuracy, with a sweet spot at $c = 0.2$. This is different from the widely used $0.08$ in the supervised learning setting. Using $0.08$ can lead to more than $1\%$ drop in accuracy compared to the optimal $0.2$ when a nonlinear projection head is applied.

**Color Jittering.** As shown in Figure 4(b), we adopt a parameter $x$ to control the strengths of color jittering function. As shown in Figure 6, increasing $x$ from $0.125$ to $2.5$ also traces a reverse-U shape, no matter whether a linear or nonlinear projection head is used. The sweet spot lies around $x = 1.0$, which is the same value as used in SimCLR [3]. Practically, we see the accuracy is more sensitive

**Figure 5:**

(a) Linear projection head  (b) MLP projection head

Figure 5: Different low-area cropping bounds in RandomResizedCrop.

**Figure 6:**

(a) Linear projection head  (b) MLP projection head

Figure 6: Different magnitudes of Color Jittering.

around the sweet spot for the nonlinear projection head, which also happens for cropping. This implies that it is important to find the sweet spot for future design of augmentation functions.

**Details.** These plots are based on the MoCo [13] framework. We use $65536$ negatives and pre-train for 100 epochs on 8 GPUs with a batch size of 256. The learning rate starts as $0.03$ and decays following a cosine annealing schedule. For the downstream task of linear evaluation, we train the linear classifier for 60 epochs with an initial learning rate of 30, following [28].

## C.3   Results on ImageNet Benchmark

Table 1: Single-crop ImageNet accuracies (%) of linear classifiers [33] trained on representations learned with different contrastive methods using ResNet-50 [15]. InfoMin Aug. refers to data augmentation using *RandomResizedCrop*, *Color Jittering*, *Gaussian Blur*, *RandAugment*, *Color Dropping*, and a *JigSaw* branch as in PIRL [23]. * indicates splitting the network into two halves.

| Method | Architecture | Param. | Head | Epochs | Top-1 | Top-5 |
|---|---|---|---|---|---|---|
| InstDis [30] | ResNet-50 | 24 | Linear | 200 | 56.5 | - |
| Local Agg. [34] | ResNet-50 | 24 | Linear | 200 | 58.8 | - |
| CMC [28] | ResNet-50* | 12 | Linear | 240 | 60.0 | 82.3 |
| MoCo [13] | ResNet-50 | 24 | Linear | 200 | 60.6 | - |
| PIRL [23] | ResNet-50 | 24 | Linear | 800 | 63.6 | - |
| CPC v2 [16] | ResNet-50 | 24 | - | - | 63.8 | 85.3 |
| SimCLR [3] | ResNet-50 | 24 | MLP | 1000 | 69.3 | 89.0 |
| InfoMin Aug. (Ours) | ResNet-50 | 24 | MLP | 200 | 70.1 | 89.4 |
| InfoMin Aug. (Ours) | ResNet-50 | 24 | MLP | 800 | **73.0** | **91.1** |

Table 2: Single-crop ImageNet accuracies (%) of linear classifiers [33] trained on representations learned with different methods using various architectures.

| Method | Architecture | Param. | Head | Epochs | Top-1 | Top-5 |
|---|---|---|---|---|---|---|
| *Methods using contrastive learning:* | | | | | | |
| InstDis [30] | ResNet-50 | 24 | Linear | 200 | 56.5 | - |
| Local Agg. [34] | ResNet-50 | 24 | Linear | 200 | 58.8 | - |
| CPC v2 [16] | ResNet-50 | 24 | - | - | 63.8 | 85.3 |
| CMC [28] | 2x ResNet-50(0.5x) | 12 | Linear | 240 | 60.0 | 82.3 |
| CMC [28] | 2x ResNet-50(1x) | 47 | Linear | 240 | 66.2 | 87.0 |
| CMC [28] | 2x ResNet-50(2x) | 188 | Linear | 240 | 70.6 | 89.7 |
| MoCo [13] | ResNet-50 | 24 | Linear | 200 | 60.6 | - |
| MoCo [13] | ResNet-50 (2x) | 94 | Linear | 200 | 65.4 | - |
| MoCo [13] | ResNet-50 (4x) | 375 | Linear | 200 | 68.6 | - |
| PIRL [23] | ResNet-50 | 24 | Linear | 800 | 63.6 | - |
| PIRL [23] | ResNet-50 (2x) | 94 | Linear | 800 | 67.4 | - |
| SimCLR [3] | ResNet-50 | 24 | MLP | 1000 | 69.3 | - |
| SimCLR [3] | ResNet-50 (2x) | 94 | MLP | 1000 | 74.2 | - |
| SimCLR [3] | ResNet-50 (4x) | 375 | MLP | 1000 | 76.5 | - |
| MoCo V2 [4] | ResNet-50 | 24 | MLP | 800 | 71.1 | - |
| InfoMin Aug. | ResNet-50 | 24 | MLP | 100 | 67.4 | 87.9 |
| InfoMin Aug. | ResNet-50 | 24 | MLP | 200 | 70.1 | 89.4 |
| InfoMin Aug. | ResNet-50 | 24 | MLP | 800 | 73.0 | 91.1 |
| InfoMin Aug. | ResNet-101 | 43 | MLP | 300 | 73.4 | - |
| InfoMin Aug. | ResNet-152 | 58 | MLP | 200 | 73.4 | - |
| InfoMin Aug. | ResNeXt-101 | 87 | MLP | 200 | 74.5 | - |
| InfoMin Aug. | ResNeXt-152 | 120 | MLP | 200 | 75.2 | - |
| *Methods NOT using contrastive learning:* | | | | | | |
| Exemplar [9, 20] | ResNet-50 (3x) | 211 | - | 35 | 46.0 | - |
| JigSaw [24, 20] | ResNet-50 (2x) | 94 | - | 35 | 44.6 | - |
| Relative Position [7, 20] | ResNet-50 (2x) | 94 | - | 35 | 51.4 | - |
| Rotation [11, 20] | RevNet-50 (4x) | 86 | - | 35 | 55.4 | - |
| BigBiGAN [8] | RevNet-50 (4x) | 86 | - | - | 61.3 | 81.9 |
| SeLa [32] | ResNet-50 | 24 | - | 400 | 61.5 | 84.0 |

On top of the "RA-CJ-Blur" augmentations shown in Figure 4, we further reduce the mutual information (or enhance the invariance) of views by using PIRL [23], i.e., adding JigSaw [24]. This improves the accuracy of the linear classifier from 63.6% to 65.9%. Replacing the widely-used linear projection head [30, 28, 13] with a 2-layer MLP [3] increases the accuracy to 67.3%. When using this nonlinear projection head, we found a larger temperature is beneficial for downstream linear readout (as also reported in [4]). All these numbers are obtained with 100 epochs of pre-training. For simplicity, we call such unsupervised pre-training as InfoMin pre-training (i.e., pre-training with our InfoMin inspired augmentation). As shown in Table 2, our InfoMin model trained with 200 epochs achieves 70.1%, outperforming SimCLR with 1000 epochs. Finally, a new state-of-the-art, 73.0% is obtained by training for 800 epochs. Compared to SimCLR requiring 128 TPUs for large batch training, our model can be trained with as less as 4 GPUs on a single machine.

For future improvement, there is still room for manually designing better data augmentation. As shown in Figure 4(a), using "RA-CJ-Blur" has not touched the sweet spot yet. Another way to is to learn to synthesize better views (augmentations) by following (and expanding) the idea of semi-supervised view learning method presented in Section 4.2.2 of the main paper.

**Different Architectures.** We further include the performance of InfoMin as well as other SoTA methods with different architectures in Table 2. Increasing the network capacity leads to significant improvement of linear readout performance on ImageNet for InfoMin, which is consistent with previous literature [28, 13, 3, 23].

Table 3: Results of object detection and instance segmentation fine-tuned on COCO. We adopt Mask R-CNN **R50-FPN**, and report the bounding box AP and mask AP on `val2017`. In the brackets are the gaps to the ImageNet supervised pre-training counterpart. For fair comparison, InstDis [30], PIRL [23], MoCo [13], and InfoMin are all pre-trained for **200** epochs.

(a) Mask R-CNN, R50-FPN, **1x** schedule

| pre-train | $AP^{bb}$ | $AP_{50}^{bb}$ | $AP_{75}^{bb}$ | $AP^{mk}$ | $AP_{50}^{mk}$ | $AP_{75}^{mk}$ |
|---|---|---|---|---|---|---|
| random init | 32.8 | 50.9 | 35.3 | 29.9 | 47.9 | 32.0 |
| supervised | 39.7 | 59.5 | 43.3 | 35.9 | 56.6 | 38.6 |
| InstDis [30] | 38.8(↓0.9) | 58.4(↓1.1) | 42.5(↓0.8) | 35.2(↓0.7) | 55.8(↓0.8) | 37.8(↓0.8) |
| PIRL [23] | 38.6(↓1.1) | 58.2(↓1.3) | 42.1(↓1.2) | 35.1(↓0.8) | 55.5(↓1.1) | 37.7(↓0.9) |
| MoCo [13] | 39.4(↓0.3) | 59.1(↓0.4) | 42.9(↓0.4) | 35.6(↓0.3) | 56.2(↓0.4) | 38.0(↓0.6) |
| InfoMin Aug. | 40.6(↑0.9) | 60.6(↑1.1) | 44.6(↑1.3) | 36.7(↑0.8) | 57.7(↑1.1) | 39.4(↑0.8) |

(b) Mask R-CNN, R50-FPN, **2x** schedule

| pre-train | $AP^{bb}$ | $AP_{50}^{bb}$ | $AP_{75}^{bb}$ | $AP^{mk}$ | $AP_{50}^{mk}$ | $AP_{75}^{mk}$ |
|---|---|---|---|---|---|---|
| random init | 38.4 | 57.5 | 42.0 | 34.7 | 54.8 | 37.2 |
| supervised | 41.6 | 61.7 | 45.3 | 37.6 | 58.7 | 40.4 |
| InstDis [30] | 41.3(↓0.3) | 61.0(↓0.7) | 45.3(↓0.0) | 37.3(↓0.3) | 58.3(↓0.4) | 39.9(↓0.5) |
| PIRL [23] | 41.2(↓0.4) | 61.2(↓0.5) | 45.2(↓0.1) | 37.4(↓0.2) | 58.5(↓0.2) | 40.3(↓0.1) |
| MoCo [13] | 41.7(↑0.1) | 61.4(↓0.3) | 45.7(↑0.4) | 37.5(↓0.1) | 58.6(↓0.1) | 40.5(↑0.1) |
| InfoMin Aug. | 42.5(↑0.9) | 62.7(↑1.0) | 46.8(↑1.5) | 38.4(↑0.8) | 59.7(↑1.0) | 41.4(↑1.0) |

## C.4 Comparing with SoTA in Transfer Learning

One goal of unsupervised pre-training is to learn transferable representations that are beneficial for downstream tasks. The rapid progress of many vision tasks in past years can be ascribed to the paradigm of fine-tuning models that are initialized from supervised pre-training on ImageNet. When transferring to PASCAL VOC [10] and COCO [22], we found our InfoMin pre-training consistently outperforms supervised pre-training as well as other unsupervised pre-training methods.

**COCO Object Detection/Segmentation.** Feature normalization has been shown to be important during fine-tuning [13]. Therefore, we fine-tune the backbone with Synchronized BN (SyncBN [25]) and add SyncBN to newly initialized layers (e.g., FPN [21]). Table 3 reports the bounding box AP and mask AP on `val2017` on COCO, using the Mask R-CNN [14] R50-FPN pipeline. All results are reported on `Detectron2` [29]. We notice that, among unsupervised approaches, only ours consistently outperforms supervised pre-training.

We have tried different popular detection frameworks with various backbones, extended the fine-tuning schedule (e.g., **6x** schedule), and compared InfoMin ResNeXt-152 [31] trained on ImageNet-1k with supervised ResNeXt-152 trained on ImageNet-5k (6 times larger than ImageNet-1k). In all cases, InfoMin consistently outperforms supervised pre-training. Please see Section D for more detailed comparisons.

**Pascal VOC Object Detection.** We strictly follow the setting introduced in [13]. Specifically, We use Faster R-CNN [26] with R50-C4 architecture. We fine-tune all layers with 24000 iterations, each consisting of 16 images.

Table 4: Pascal VOC object detection. All contrastive models are pretrained for **200** epochs on ImageNet for fair comparison. We use Faster R-CNN R50-**C4** architecture for object detection. APs are reported using the average of 5 runs. * we use numbers from [13] since the setting is exactly the same.

| pre-train | $AP_{50}$ | AP | $AP_{75}$ | ImageNet Acc(%) |
|---|---|---|---|---|
| random init.* | 60.2 | 33.8 | 33.1 | - |
| supervised* | 81.3 | 53.5 | 58.8 | 76.1 |
| InstDis | 80.9 | 55.2 | 61.2 | 59.5 |
| PIRL | 81.0 | 55.5 | 61.3 | 61.7 |
| MoCo* | 81.5 | 55.9 | 62.6 | 60.6 |
| InfoMin Aug. (ours) | **82.7** | **57.6** | **64.6** | **70.1** |

# D  Transfer Learning with Various Backbones and Detectors on COCO

We evaluated the transferability of various models pre-trained with InfoMin, under different detection frameworks and fine-tuning schedules. In **all** cases we tested, models pre-trained with InfoMin outperform those pre-trained with supervised cross-entropy loss. Interestingly, ResNeXt-152 trained with InfoMin on **ImageNet-1K** beats its supervised counterpart trained on **ImageNet 5K**, which is **6x** times larger. Bounding box AP and mask Ap are reported on `val2017`

## D.1  ResNet-50 with Mask R-CNN, C4 architecture

The results of Mask R-CNN with R-50 C4 backbone are shown in Table 5. We experimented with **1x** and **2x** schedule.

Table 5: COCO object detection and instance segmentation. **R50-C4**. In the brackets are the gaps to the ImageNet supervised pre-training counterpart. In green are gaps of $\geq 0.5$ point. * numbers are from [13] since we use exactly the same fine-tuning setting.

(a) Mask R-CNN, R50-**C4**, **1x** schedule

| pre-train | $AP^{bb}$ | $AP^{bb}_{50}$ | $AP^{bb}_{75}$ | $AP^{mk}$ | $AP^{mk}_{50}$ | $AP^{mk}_{75}$ |
|---|---|---|---|---|---|---|
| random init* | 26.4 | 44.0 | 27.8 | 29.3 | 46.9 | 30.8 |
| supervised* | 38.2 | 58.2 | 41.2 | 33.3 | 54.7 | 35.2 |
| MoCo* | 38.5(↑0.3) | 58.3(↑0.1) | 41.6(↑0.4) | 33.6(↑0.1) | 54.8(↑0.1) | 35.6(↑0.1) |
| InfoMin Aug. | 39.0(↑0.8) | 58.5(↑0.3) | 42.0(↑0.8) | 34.1(↑0.8) | 55.2(↑0.5) | 36.3(↑1.1) |

(b) Mask R-CNN, R50-**C4**, **2x** schedule

| pre-train | $AP^{bb}$ | $AP^{bb}_{50}$ | $AP^{bb}_{75}$ | $AP^{mk}$ | $AP^{mk}_{50}$ | $AP^{mk}_{75}$ |
|---|---|---|---|---|---|---|
| random init* | 35.6 | 54.6 | 38.2 | 31.4 | 51.5 | 33.5 |
| supervised* | 40.0 | 59.9 | 43.1 | 34.7 | 56.5 | 36.9 |
| MoCo* | 40.7(↑0.7) | 60.5(↓0.6) | 44.1(↑1.0) | 35.6(↓0.7) | 57.4(↓0.8) | 38.1(↑0.7) |
| InfoMin Aug. | 41.3(↑1.3) | 61.2(↑1.3) | 45.0(↑1.9) | 36.0(↑1.3) | 57.9(↑1.4) | 38.3(↑1.4) |

## D.2  ResNet-50 with Mask R-CNN, FPN architecture

The results of Mask R-CNN with R-50 FPN backbone are shown in Table 6. We compared with MoCo [13] and MoCo v2 [4] under **2x** schedule, and also experimented with **6x** schedule.

Table 6: COCO object detection and instance segmentation. **R50-FPN**. In the brackets are the gaps to the ImageNet supervised pre-training counterpart. In green are gaps of $\geq 0.5$ point.

(a) Mask R-CNN, R50-**FPN**, **2x** schedule

| pre-train | $AP^{bb}$ | $AP^{bb}_{50}$ | $AP^{bb}_{75}$ | $AP^{mk}$ | $AP^{mk}_{50}$ | $AP^{mk}_{75}$ |
|---|---|---|---|---|---|---|
| random init | 38.4 | 57.5 | 42.0 | 34.7 | 54.8 | 37.2 |
| supervised | 41.6 | 61.7 | 45.3 | 37.6 | 58.7 | 40.4 |
| MoCo [13] | 41.7(↑0.1) | 61.4(↓0.3) | 45.7(↑0.4) | 37.5(↓0.1) | 58.6(↓0.1) | 40.5(↑0.1) |
| MoCo v2 [4] | 41.7(↑0.1) | 61.6(↓0.1) | 45.6(↑0.3) | 37.6(↓0.0) | 58.7(↓0.0) | 40.5(↑0.1) |
| InfoMin Aug. | 42.5(↑0.9) | 62.7(↑1.0) | 46.8(↑1.5) | 38.4(↑0.8) | 59.7(↑1.0) | 41.4(↑1.0) |

(b) Mask R-CNN, R50-**FPN**, **6x** schedule

| pre-train | $AP^{bb}$ | $AP^{bb}_{50}$ | $AP^{bb}_{75}$ | $AP^{mk}$ | $AP^{mk}_{50}$ | $AP^{mk}_{75}$ |
|---|---|---|---|---|---|---|
| random init | 42.7 | 62.6 | 46.7 | 38.6 | 59.9 | 41.6 |
| supervised | 42.6 | 62.4 | 46.5 | 38.5 | 59.9 | 41.5 |
| InfoMin Aug. | 43.6(↑1.0) | 63.6(↑1.2) | 47.3(↑0.8) | 39.2(↑0.7) | 60.6(↑0.7) | 42.3(↑0.8) |

## D.3  ResNet-101 with Mask R-CNN, C4 architecture

The results of Mask R-CNN with R-101 C4 backbone are shown in Table 7. We experimented with **1x** and **1x** schedule.

Table 7: COCO object detection and instance segmentation. **R101-C4**. In the brackets are the gaps to the ImageNet supervised pre-training counterpart.

(a) Mask R-CNN, R101-**C4**, **1x** schedule

| pre-train | $AP^{bb}$ | $AP^{bb}_{50}$ | $AP^{bb}_{75}$ | $AP^{mk}$ | $AP^{mk}_{50}$ | $AP^{mk}_{75}$ |
|---|---|---|---|---|---|---|
| supervised | 40.9 | 60.6 | 44.2 | 35.1 | 56.8 | 37.3 |
| InfoMin Aug. | 42.5(↑1.6) | 62.1(↑1.5) | 46.1(↑1.9) | 36.7(↑1.6) | 58.7(↑1.9) | 39.2(↑1.9) |

(b) Mask R-CNN, R101-**C4**, **2x** schedule

| pre-train | $AP^{bb}$ | $AP^{bb}_{50}$ | $AP^{bb}_{75}$ | $AP^{mk}$ | $AP^{mk}_{50}$ | $AP^{mk}_{75}$ |
|---|---|---|---|---|---|---|
| supervised | 42.5 | 62.3 | 46.1 | 36.4 | 58.7 | 38.7 |
| InfoMin Aug. | 43.9(↑1.4) | 63.5(↑1.2) | 47.5(↑1.4) | 37.8(↑1.4) | 60.4(↑1.7) | 40.2(↑1.5) |

## D.4  ResNet-101 with Mask R-CNN, FPN architecture

The results of Mask R-CNN with R-101 FPN backbone are shown in Table 8. We experimented with **1x**, **2x**, and **6x** schedule.

Table 8: COCO object detection and instance segmentation. **R101-FPN**. In the brackets are the gaps to the ImageNet supervised pre-training counterpart.

(a) Mask R-CNN, R101-**FPN**, **1x** schedule

| pre-train | $AP^{bb}$ | $AP^{bb}_{50}$ | $AP^{bb}_{75}$ | $AP^{mk}$ | $AP^{mk}_{50}$ | $AP^{mk}_{75}$ |
|---|---|---|---|---|---|---|
| supervised | 42.0 | 62.3 | 46.0 | 37.6 | 59.1 | 40.1 |
| InfoMin Aug. | 42.9(↑0.9) | 62.6(↑0.3) | 47.2(↑1.2) | 38.6(↑1.0) | 59.7(↑0.6) | 41.6(↑1.5) |

(b) Mask R-CNN, R101-**FPN**, **2x** schedule

| pre-train | $AP^{bb}$ | $AP^{bb}_{50}$ | $AP^{bb}_{75}$ | $AP^{mk}$ | $AP^{mk}_{50}$ | $AP^{mk}_{75}$ |
|---|---|---|---|---|---|---|
| supervised | 43.3 | 63.3 | 47.1 | 38.8 | 60.1 | 42.1 |
| InfoMin Aug. | 44.5(↑1.2) | 64.4(↑1.1) | 48.8(↑1.7) | 39.9(↑1.1) | 61.5(↑1.4) | 42.9(↑0.8) |

(c) Mask R-CNN, R101-**FPN**, **6x** schedule

| pre-train | $AP^{bb}$ | $AP^{bb}_{50}$ | $AP^{bb}_{75}$ | $AP^{mk}$ | $AP^{mk}_{50}$ | $AP^{mk}_{75}$ |
|---|---|---|---|---|---|---|
| supervised | 44.1 | 63.7 | 48.0 | 39.5 | 61.0 | 42.4 |
| InfoMin Aug. | 45.3(↑1.2) | 65.0(↑1.3) | 49.3(↑1.3) | 40.5(↑1.0) | 62.5(↑1.5) | 43.7(↑1.3) |

## D.5  ResNet-101 with Cascade Mask R-CNN, FPN architecture

The results of Cascade [2] Mask R-CNN with R-101 FPN backbone are shown in Table 9. We experimented with **1x**, **2x**, and **6x** schedule.

## D.6  ResNeXt-101 with Mask R-CNN, FPN architecture

The results of Mask R-CNN with X-101 FPN backbone are shown in Table 10. We experimented with **1x** and **2x** schedule.

Table 9: COCO object detection and instance segmentation. **Cascade R101-FPN**. In the brackets are the gaps to the ImageNet supervised pre-training counterpart.

(a) Cascade Mask R-CNN, R101-**FPN**, **1x** schedule

| pre-train | $AP^{bb}$ | $AP^{bb}_{50}$ | $AP^{bb}_{75}$ | $AP^{mk}$ | $AP^{mk}_{50}$ | $AP^{mk}_{75}$ |
|---|---|---|---|---|---|---|
| supervised | 44.9 | 62.3 | 48.8 | 38.8 | 59.9 | 42.0 |
| InfoMin Aug. | 45.8(↑0.9) | 63.1(↑0.8) | 49.5(↑0.7) | 39.6(↑0.8) | 60.4(↑0.5) | 42.9(↑0.9) |

(b) Cascade Mask R-CNN, R101-**FPN**, **2x** schedule

| pre-train | $AP^{bb}$ | $AP^{bb}_{50}$ | $AP^{bb}_{75}$ | $AP^{mk}$ | $AP^{mk}_{50}$ | $AP^{mk}_{75}$ |
|---|---|---|---|---|---|---|
| supervised | 45.9 | 63.4 | 49.7 | 39.8 | 60.9 | 43.0 |
| InfoMin Aug. | 47.3(↑1.4) | 64.6(↑1.2) | 51.5(↑1.8) | 40.9(↑1.1) | 62.1(↑1.2) | 44.6(↑1.6) |

(c) Cascade Mask R-CNN, R101-**FPN**, **6x** schedule

| pre-train | $AP^{bb}$ | $AP^{bb}_{50}$ | $AP^{bb}_{75}$ | $AP^{mk}$ | $AP^{mk}_{50}$ | $AP^{mk}_{75}$ |
|---|---|---|---|---|---|---|
| supervised | 46.6 | 64.0 | 50.6 | 40.5 | 61.9 | 44.1 |
| InfoMin Aug. | 48.2(↑1.6) | 65.8(↑1.8) | 52.7(↑2.1) | 41.8(↑1.3) | 63.5(↑1.6) | 45.6(↑1.5) |

Table 10: COCO object detection and instance segmentation. **X101-FPN**. In the brackets are the gaps to the ImageNet supervised pre-training counterpart.

(a) Mask R-CNN, X101-**FPN**, **1x** schedule

| pre-train | $AP^{bb}$ | $AP^{bb}_{50}$ | $AP^{bb}_{75}$ | $AP^{mk}$ | $AP^{mk}_{50}$ | $AP^{mk}_{75}$ |
|---|---|---|---|---|---|---|
| supervised | 44.1 | 64.8 | 48.3 | 39.3 | 61.5 | 42.3 |
| InfoMin Aug. | 45.0(↑0.9) | 65.3(↑0.5) | 49.5(↑1.2) | 40.1(↑0.8) | 62.3(↑0.8) | 43.1(↑0.8) |

(b) Mask R-CNN, X101-**FPN**, **2x** schedule

| pre-train | $AP^{bb}$ | $AP^{bb}_{50}$ | $AP^{bb}_{75}$ | $AP^{mk}$ | $AP^{mk}_{50}$ | $AP^{mk}_{75}$ |
|---|---|---|---|---|---|---|
| supervised | 44.6 | 64.4 | 49.0 | 39.8 | 61.6 | 43.0 |
| InfoMin Aug. | 45.4(↑0.8) | 65.3(↑0.9) | 49.6(↑0.6) | 40.5(↑0.7) | 62.5(↑0.9) | 43.8(↑0.8) |

## D.7 ResNeXt-152 with Mask R-CNN, FPN architecture

The results of Mask R-CNN with X-152 FPN backbone are shown in Table 11. We experimented with **1x** schedule.. Note in this case, while InfoMin model is pre-trained on the standard ImageNet-1K dataset, supervised model is pre-trained on ImageNet-5K, which is **6x** times larger than ImageNet-1K. That said, we found InfoMin still outperforms the supervised pre-training.

Table 11: COCO object detection and instance segmentation. **X152-FPN**. In the brackets are the gaps to the ImageNet supervised pre-training counterpart. Supervised model is pre-trained on ImageNet-5K, while InfoMin model is only pre-trained on ImageNet-1K.

(a) Mask R-CNN, X152-**FPN**, **1x** schedule

| pre-train | $AP^{bb}$ | $AP^{bb}_{50}$ | $AP^{bb}_{75}$ | $AP^{mk}$ | $AP^{mk}_{50}$ | $AP^{mk}_{75}$ |
|---|---|---|---|---|---|---|
| supervised | 45.6 | 65.7 | 50.1 | 40.6 | 63.0 | 43.5 |
| InfoMin Aug. | 46.4(↑0.8) | 66.5(↑0.8) | 50.8(↑0.7) | 41.3(↑0.7) | 63.6(↑0.6) | 44.4(↑0.9) |

## Footnotes

*Project page: http://hobbitlong.github.io/InfoMin