[Reviews · NeurIPS 2020]

Review 1

Summary and Contributions: This paper first proposed a theoretic analysis from the perspective of mutual information to define what are the good augmentations for contrastive learning and whether the augmentation is to excessive for the downstream tasks. Preliminary experiments on toy examples show that there is a sweet spot for the augmentation used in contrastive learning. Since such sweet spot is highly correlated with the downstream tasks, the author proposed a semi-supervised learning scenario to search for good augmentations. Extensive experiments support the theoretical claim and demonstrate that a good augmentation could improve the performance.

Strengths: • The entire paper is well written and the idea is clearly presented. • The analysis of a good choice of view selection is interesting and could potentially provide a good guideline for the following research about view selection. • The theoretical claim is sound and thus provides a new perspective of viewing the contrastive learning problem. • Both Figure1 and Figure2 provide a good illustration of the idea in the paper. • Extensive experiments validate the hypothesis that there is a sweet spot for view selection.

Weaknesses: • In line 121, the optimal z* is derived based on the fact that the downstream task is known (Line 145-146). However, for a pretrained model that can served for multiple downstream tasks, the optimal z* for one of the tasks might not be closed to optimal for another task. In other word, the z* selected from the proposed method might have worse generalizability than the z learning by standard contrastive learning. A clear example of this is the result from Table 1. To demonstrate the fact that optimal views depend on the downstream task, the author constructs a toy experiments as shown in Table 1. This experiment exposes the fact that when only selecting specific views for specific downstream task, the learned representation cannot generalize well to other tasks. For example, when the task shifts from digit classification to background classification, the performance of pretrained model that relies on digits information becomes the worse on background classification. This shows that an optimal z* for one task is unable to generalize well to another task. In addition, the author should compare to the baseline that contains all factors (i.e. digit, bkdg and pose) in Table 1. One can do this without knowing what the downstream tasks are in advance, which can potentially make the model more generalizable. • Moreover, given that the author proposed a new manner to perform semi-supervised learning, baselines in semi-supervised learning literature should be compared. It would be interesting to show that the proposed method can outperform other semi-supervised methods. Furthermore, the goal in self-supervised training is to enable the pretrained model to serve for many downstream tasks. However, it is unclear that the g function learned from semi-supervised loss of Eq3 will enable the pretrained model to adopt to different downstream tasks. The author is suggested to discuss whether g will overfit to a specific downstream task. • In figure2, the transfer performance is only focused on single task. Moreover, it seems most of the examples (i.e. Figure 3, 5 and 6) that the author shows follow the case from Figure 2(a) (i.e. reversed U shape). What are some examples for Figure 2(b)? • As mentioned in Line 216-219 and Line 119-125 in Appendix, it seems that the author is searching for the best augmentation sets to achieve the best ImageNet performance. There is in fact no technical contribution on this procedure (Line 209-214). Given enough computation resources, one can simply try all the possible augmentation techniques and achieve better accuracy for the downstream task. • Line 273. The author mentioned that the GAN style training in Eq2 is unstable. Does the conclusion in Figure 6(a) always hold in different runs? The author is suggested to show the variations between runs. • In Table 2, is the supervised baseline trained only on few data that has labeled? • Line 194-203. It is a little misleading from the text that the performance gain of prior works [52, 37, 7, 41] is mainly due to smaller mutual information of selected views. For example, the sentence “SimCLR uses a stronger class of augmentations T’, which leads to smaller mutual information between the two views than InstDis”. As the author claims, there is a sweet spot for the data augmentation, so stronger augmentations does not necessary lead to better performance. Post rebuttal: Thanks the author for providing a constructive rebuttal. After reading the review and the rebuttal, I would tend to accept the submission for its systematic study of the reverse-U shape phenomenon. The author is suggested to add more discussion about the generator g as suggested by most reviewers.

Correctness: The discussion in section 3 is sound. However, there are some claims that are kind of misleading. Please refer to the weakness section.

Clarity: Yes, this paper is well written. Most the ideas and experimental detail are clearly presented.

Relation to Prior Work: This paper is related to the literature of self-supervised learning and semi-supervised learning. The related work section for the self-supervised learning literature is complete and covers most of the recent self-supervised learning works. However, this paper does not compare to semi-supervised learning literature. For example, [60] could a good baseline for demonstrating effectiveness of the proposed InfoMin augmentation.

Reproducibility: Yes

Additional Feedback: The author is suggested to clarify the confusion in the weakness section.


Review 2

Summary and Contributions: This paper studies contrastive learning methods for self-supervised representation learning. It studies how multiple views of the same data are used for representation learning, and how the mutual information between these views matters for downstream performance. The authors propose a theory that there is a sweet spot in the amount of mutual information between two views (not too less, not too much) such that the downstream performance is highest at this point. They empirically verify this theory for two classes of views (patches, and colors). They propose a method that simply combines existing augmentations from prior work and provides gains over them.

Strengths: 1. While there are many papers now which talk about using views for representation learning, there is little work on understanding why certain views work. This paper is a step in this direction and is quite relevant to self-supervised learning practioners. 2. The empirical observations in Figure 3 are really interesting. There is a strong correlation between I_NCE and the final performance and the authors study control variables like patch distance and color space to further make this effect interesting to study. 3. Section 4 is full of interesting analysis (a). 4.1 shows how the views for pretraining are dependent on the final downstream task itself. (b). 4.2.2 shows how one can generate views by using some information from the downstream task. This can be viewed as a form of learned data augmentation and is a really interesting direction.

Weaknesses: . The currently proposed theory is quite limited and provides limited actionable insights for developing new techniques. Here are a few important questions left largely unanswered -- (a) Why do certain views work better than others? For example, why does L + ab work worse than the image + patch view? Is there a way to quantify this. (b) While the authors empirically verify the schematic in Figure 1(c), it does little to inform us how to design the views without training models. Since I_NCE is computed using the model trained on the views, this measure still requires one to train models. Since the model is already trained, one may also directly compute the transfer performance. 2. I_NCE is essentially -L_NCE which means that Figures 3, 4, 5 really show a correlation between the transfer performance and the training loss. InstDisc [57] also shows such correlations (although only in the limited context of instance discrimination and not studying variables like patch distance) and it would be good to note this. 3. The proposed InfoMin Aug is not explored in great detail. This result in mentioned in the abstract, but the paper spend one paragraph (L209) to explain everything about this method - the model, the data, the training, the transfer result. The authors mention that this method uses augmentation from prior work, but it is not clear how these are combined. Also, since each of these augmentations provides a different type of view, it would be good to quantify how much each augmentation matters. Questions q1 - How is g parametrized in 4.2.1? q2 - Isn't Figure 2 also a schematic? Might be good to clarify that as a hypothesis/schematic, rather than an observation.

Correctness: I believe they are (the authors do conflate correlation with causation, but that's a minor nit).

Clarity: Yes, it is well written and was an enjoyable read. The experimental setup in this work is not quite clear, which makes it a little hard to compare these results with those from other prior work. But since the main point of this paper is to study views empirically, this is does not lessen its contribution. I hope the authors release their code and models from Figures 3-5.

Relation to Prior Work: Yes.

Reproducibility: No

Additional Feedback:


Review 3

Summary and Contributions: This paper investigates what makes for good views for contrastive learning from a perspective of mutual information. An important conclusion, validated by adequate experiments, is that there exists a sweet spot for model generalization where the information shared by views are sufficient for the task and does not contain other noise. Authors also propose a simple adversarial method to learn good views.

Strengths: + The paper is well motivated and well written. This paper poses a good question: How to construct good views for contrastive learning? What are the properties of the good views? + The experiments are well organized to validate hypotheses properly. + The idea of learning good views for contrastive learning is interesting. + ImageNet results are promising (~+4% than SimCLR)

Weaknesses: - I highly value the problem posed , but the core finding in this paper "transfer performance will be upper-bounded by a reverse-U shaped curve, with the sweet spot at the top of the cure" seems trivial to me. In all the experiments the authors use I_{NCE} to represent the mutual information, and show reverse-U shape transfer performance curves w.r.t I_{NCE}. In my opinion, this is equivalent to show a reverse-U shape curve w.r.t the training loss, and this is trivial because the training undergoes under-fitting, critical-fitting and over-fitting stages. However I still appreciate the authors conduct thorough experiments to show this finding. - The problem of finding the optimal views is not well-solved. Authors try to learn optimal views via adversirial training, but the solution now is kind of naive yet. For example, the views are defined as linear transformation (via learnable 1x1 conv) only, but more complicated view construction method is not explored. - Also, it is not convicing that learned optimal views are not superior to YDbDr composition. Maybe the learning method is not good enough, or the search space is not well designed. - The last contribution of this paper, "Applying our understanding to achieve state of the art accuracy of 73.0% on the ImageNet linear readout benchmark with a ResNet-50", seems weak. The performance is good, but what the authors do is only strengthening the data augmentation function. It is expected the authors design a method to find the optimal views (a sweet spot), but they merely decrease the mutual information between views. Therefore I think this good result does not well-support the core idea of this paper. - A suggestion: According to my understanding, what the authors trying to say in Section 3.4 is that they can unify recent contrastive learning methods by the InfoMin principle. I think it may be better if the authors spend less space on this section, and instead highlight they own ImageNet results, because unifying the recent methods is not a contribution claimed, and less relevant to the main idea. - Why curves in Figure 4 is not reverse-U shaped? Will they be reverse-U shaped if keep reducing the mutual information? Post reubuttal: I have read the authors' response and raise my rating to a "7-accept", because most my concerns are addressed and some my mis-understandings are carefully expainled.

Correctness: Yes.

Clarity: Writting is good.

Relation to Prior Work: Yes

Reproducibility: Yes

Additional Feedback: Overall I think this is a solid paper, I think it would be more rounded if my suggestions are considered.


Review 4

Summary and Contributions: This paper shows the impact of view selection in contrastive learning. In particular, it demonstrates the optimal view selection depends on the downstream tasks and the ideal selected views should share minimal mutual information required for the downstream task and throw away all other information.

Strengths: 1- Paper shows the mutual information between views (for contrastive learning) and the accuracy obtained for the downstream tasks, have usually a reversed U-shape relationship meaning if the mutual information is too low or too high, the accuracy on the downstream task is deteriorated due to either lack of sufficient information (low mutual information) or presence of noise (high mutual information). There is a sweet spot in-between, where the mutual information is optimal for the downstream task, the necessary information is maintained and the noise is thrown away. 2- The authors demonstrate in a well-designed experiment (Section 4.1) that the optimal views depend on the downstream task. This experiment demonstrates nicely the impact of the selected views to either maintain relevant information or obtained undesired noisy information. 3- Authors propose a generative process, where two views are generated in a way that they share minimal mutual information between them and meanwhile maintain the relevant information for the downstream task through using a semi-supervised learning approach, in which the labels (of the downstream task) are used on some percentage of data to maintain the vital information in the generated views. 4- Authors show by using multiple data-augmentations together (and hence reducing the mutual information in-between views) they can obtain SOTA on unsupervised ImageNet classification and later on the downstream tasks by using the features learned from an ImageNet pre-trained model.

Weaknesses: While I think the points 1 and 2 mentioned above are very interesting observations and would benefit the community, the main flaw I found was in the generative process of learning views in the unsupervised setting. In particular, regarding the model presented in Section 4.2.1, what stops g from learning trivial solutions? For example, g can generate a \hat{X}, which can be even unrelated to X and have no shared information between channels 1 and 2,3. So the mutual information would be minimized and no relevant information of the image would be maintained. When this approach is used in the semi-supervised setups (by using labels), the model maintains the information required for the downstream tasks, but I am surprised that the model can work in the unsupervised setting. The fact that f1 and f2 maximize the mutual information is irrelevant here, as f1 and f2 are secondary steps after function g provides \hat{X}, so g can make the information in \hat{X} even unrelated to X to minimize its loss regardless of what g aims to do. Why such trivial solutions do not happen? Regarding the experiment in Section 4.1 and Supp. B.4, the authors use 10 frames for view 1 and one frame for view 2. When the downstream task is digit position, the second view shares a similar location of digit compared to which frame in view 1? Since the goal is to find the digit location later in the downstream task, isn’t using multiple frames (with a moving digit) instead of one frame in view 1 (that has the same digit location) contradictory to the goal the authors are pursuing and gives the model irrelevant information on the digit location?

Correctness: In general yes, except the ones mentioned above.

Clarity: Yes

Relation to Prior Work: Yes

Reproducibility: No

Additional Feedback: Although a lot of information is provided by authors about the implementation details, I think the results would not be fully reproducible without providing the code. So, I advise authors to release the code to the community.

[Author Response · NeurIPS 2020]

We thank all reviewers for their constructive comments and feedback. We have already provided the code in the supplementary material, and will open source it upon acceptance.

To Reviewer 1: **(1) "Optimal $z^*$ is task dependent".** This is not a weakness of our approach, but a point that we emphasize through our analysis in the colored moving MNIST experiments. It indicates that we cannot find a pair of views that are universally optimal for all downstream tasks. The baseline you suggest is included in Table 1 of the supplement, which shows that when all factors (digit, bkdg and pose) are used to create views, the learned $z$ only works well for *background* classification, but does not help *digit classification* and *localization*. This shows that $z$ learned without view selection is not as generalizable as we might think. **(2) "Semi-supervised baselines".** Our focus is on semi-supervised *view* (not feature) learning for verifying the InfoMin hypothesis and supporting our analysis, not to achieve SOTA semi-supervised feature learning. While our contrastive feature learning stage given learned views is unsupervised, we achieved comparable performance as the SOTA semi-supervised learning methods (e.g., on STL-10, our method achieves 5.75% error rate, while MixMatch obtains 5.59%). In the future, our semi-supervised view-learning algorithm could be combined with semi-supervised contrastive representation learning algorithms to further improve performance. **(3) "whether $g$ overfits to a specific task".** The main purpose of learning $g$ with a semi-supervised loss is to verify our InfoMin hypothesis. In theory, it is possible that $g$ makes pre-trained models perform better on tasks similar to the supervised task used to train $g$, but worse on less similar tasks. We will expand our discussion as suggested. **(4) Figure 2**. Figure 2 is schematic; what are signals and nuisances depends on the downstream task, e.g, signals for one task might be nuisances for another. Empirically we have only observed behavior as depicted Figure 2(a), but in theory Figure 2(b) could also happen. **(5) "No technical contribution on ImageNet augmentation".** Our main goal was to analyze the reverse-U shape phenomenon on a larger-scale and practical data augmentation setup, not to propose new techniques for data augmentation. **(6) "variations between runs for GAN-style training".** Figure 6(a) already includes multiple runs. There is instability in the sense that each single run might end up with a different amount of MI, but the trend of reverse-U shape between MI and accuracy with multiple runs is stable. **(7) "Supervised baseline in Table 2".** Yes, it is trained only on the labeled subset. We will rename the items to make it clearer. **(8) Augmentation in SimCLR** has *not* reached the sweet spot yet. See 'CJ-Blur' (which is SimCLR augmentation) in Figure 4(a) in the supplement.

To Reviewer 2: **(1) "L+ab v.s. image+patch"**. These two setups are not directly comparable since "image+patch" is trained on a different dataset, and please see Sec B.1 in supplement for the reason. Generally, as shown in Table 1, certain views will work better if the shared factors between views are related to the downstream task, as highlighted in our toy MNIST experiments. **(2) "usage of schematic in Figure 1(c)".** One way of making this scheme more practical would be to compute $I_{NCE}$ on smaller models first, or train on a subset of the data to identify good views. This direction deserves further study in future works. **(3) "Correlation of $L_{NCE}$ and downstream accuracy in InstDis".** Thanks for pointing this out, we will note this. We have also clarified in the text that $I_{NCE}$ refers to the *converged* loss, rather than *unconverged* loss along the training. **(4) "how much each augmentation matters".** This is presented in Fig 4 of supplement. We will modify L209 accordingly. **(5) "how $g$ is parameterized".** $g$ consists of several blocks, each with several 1x1 convolutions and relu activations. See B.5 in Supp for more details. **(6) "Figure 2 a schematic?"**. Yes and we will make it clearer in revised version.

To Reviewer 3: **(1) "reverse-U shape corresponds to under-fitting, critical-fitting and over-fitting stages".** This is not true. $I_{NCE}$ in our paper means *converged* loss, not the loss during the training procedure. For each plot, we only vary input views ($\mathbf{v}_1$ and $\mathbf{v}_2$) and train until convergence to get $I_{NCE}$. We also evaluated $I_{NCE}$ on held-out *validation* data, showing an almost identical reverse-U shape. **(2) "views are defined as linear transformation".** The learned views are more complex: $g$ is a stack of multiple blocks, each consisting of 1x1 convolutions and 'relu' non-linearity (see B.5 and code for details). Note that the view learning experiments is mainly for verifying InfoMin hypothesis. It is still preliminary but is an interesting future direction. **(3) "reasons for not superior to YDbDr composition".** We agree with the explanations R3 provided and leave it as future work. **(4) "Analysis and results on ImageNet".** The core idea of this paper is the InfoMin principle, and its derivative – reverse-U shape. The augmentation analysis on ImageNet is mainly used to support this hypothesis (see Fig 5 in main paper and Fig 4-5 in Supp). We will modify the last contribution accordingly. 'How to learn optimal views (or augmentations)' is an interesting direction but not the primary focus in this paper. We will adopt the suggestion about rewriting Sec. 3.4. **(5) "curves in Figure 4 is not reverse-U shaped?".** This is because there is no other natural color spaces that further reduce MI. So we used learning methods to synthesize neural color spaces with less MI, as shown in Sec 4.2. If you combine the results in Figure 4(a) with Figure 6(a), you will observe the reverse-U shape.

To Reviewer 4: **(1) "trivial solution of $g(\cdot)$".** We avoid such trivial solution by constraining $g(\cdot)$ to be an invertible function, similar to flow-based generative models. Therefore, $g(\cdot)$ is a bijective mapping and total information is preserved after the transformation $g(\cdot)$. **(2) "second view shares a similar location of digit compared to which frame in view 1?"** Given a sequence $\mathbf{x}_{1:20}$, we use the first 10 frames $\mathbf{x}_{1:10}$ as $\mathbf{v}_1$, the digit position of $\mathbf{v}_2$ is the same as the 20-th frame of $\mathbf{x}$, i.e., $\mathbf{x}_{20}$. Therefore, contrastive learning requires the model to extract the position of digits in all 10 frames of $\mathbf{v}_1$ and then extrapolate the motion to predict the digit position in $\mathbf{v}_2$. All position information in $\mathbf{v}_1$ is thus relevant to that in $\mathbf{v}_2$.

[Meta-Review · NeurIPS 2020]

The paper studies contrastive methods for self-supervised representation learning. It studies how multiple views of the same data are used for representation learning, and how the mutual information between these views matters for downstream performance. The authors propose a theory that there is a sweet spot in the amount of mutual information between two views (not too less, not too much) such that the downstream performance is highest at this point. They empirically verify this theory for two classes of views (patches, and colors). They propose a method that simply combines existing augmentations from prior work and provides gains over them. The paper was reviewed by four reviewers that were all positive and recommended acceptance. The reviewers liked the theoretical basis of the paper and the resulting insight about the sweet spot for model generalization, as well as the experiments presented to validate the theory. The reviewers raised many technical questions, which were satisfactorily addressed by the author rebuttal. In terms of weaknesses, some reviewers noted that the proposed theory provides limited actionable insights for developing new techniques, and that the paper has limited technical contribution in terms of algorithmic advances to contrastive learning. These limitations weaken its impact. Nevertheless, there was consensus that the paper deserves publication.